Corrected: Author correction

# Variants in exons 5 and 6 of *ACTB* cause syndromic thrombocytopenia

Sharissa L. Latham [1], Nadja Ehmke[2,3], Patrick Y.A. Reinke[1,4], Manuel H. Taft [1], Dorothee Eicke[5], Theresia Reindl[1], Werner Stenzel[6], Michael J. Lyons[7], Michael J. Friez[7], Jennifer A. Lee[7], Ramona Hecker[8], Michael C. Frühwald[9], Kerstin Becker[10], Teresa M. Neuhann[10], Denise Horn[2], Evelin Schrock[11], Indra Niehaus[11], Katharina Sarnow[11], Konrad Grützmann[12], Luzie Gawehn[11], Barbara Klink [11], Andreas Rump[11], Christine Chaponnier[13], Constanca Figueiredo[5], Ralf Knöfler[14], Dietmar J. Manstein[1,4] & Nataliya Di Donato [11]

Germline mutations in the ubiquitously expressed *ACTB*, which encodes β-cytoplasmic actin (CYA), are almost exclusively associated with Baraitser-Winter Cerebrofrontofacial syndrome (BWCFF). Here, we report six patients with previously undescribed heterozygous variants clustered in the 3′-coding region of *ACTB*. Patients present with clinical features distinct from BWCFF, including mild developmental disability, microcephaly, and thrombocytopenia with platelet anisotropy. Using patient-derived fibroblasts, we demonstrate cohort specific changes to β-CYA filament populations, which include the enhanced recruitment of thrombocytopenia-associated actin binding proteins (ABPs). These perturbed interactions are supported by in silico modeling and are validated in disease-relevant thrombocytes. Co-examination of actin and microtubule cytoskeleton constituents in patient-derived megakaryocytes and thrombocytes indicates that these β-CYA mutations inhibit the final stages of platelet maturation by compromising microtubule organization. Our results define an *ACTB*-associated clinical syndrome with a distinct genotype-phenotype correlation and delineate molecular mechanisms underlying thrombocytopenia in this patient cohort.

[1] Institute for Biophysical Chemistry, Hannover Medical School, Hannover 30625, Germany. [2] Institute of Medical and Human Genetics, Charité-Universitätsmedizin Berlin, Berlin 13353, Germany. [3] Berlin Institute of Health, Berlin 10117, Germany. [4] Division for Structural Biochemistry, Hannover Medical School, Hannover 30625, Germany. [5] Institute for Transfusion Medicine, Hannover Medical School, Hannover 30625, Germany. [6] Department of Neuropathology, Charité-Universitätsmedizin Berlin, Berlin 10117, Germany. [7] Greenwood Genetic Center, Greenwood, South Carolina, SC 29646, USA. [8] Institute for Clinical Chemistry and Laboratory Medicine, Medical Faculty of TU Dresden, Dresden 01307, Germany. [9] Swabian Children's Cancer Center, Children's Hospital Augsburg, Augsburg 86156, Germany. [10] Medical Genetics Center, Munich 80335, Germany. [11] Institute for Clinical Genetics, TU Dresden, Dresden 01307, Germany. [12] Core Unit for Molecular Tumor Diagnostics, National Center for Tumor Diseases Dresden, Dresden 01307, Germany. [13] Department of Pathology-Immunology, Faculty of Medicine, University of Geneva, Geneva 1211, Switzerland. [14] Department of Paediatric Haemostaseology, Medical Faculty of TU Dresden, Dresden 01307, Germany. These authors contributed equally: Sharissa L. Latham, Nadja Ehmke. Correspondence and requests for materials should be addressed to S.L.L. (email: Latham.Sharissa@mh-hannover.de) or to D.J.M. (email: Manstein.Dietmar@mh-hannover.de) or to N.DD. (email: Nataliya.didonato@uniklinikum-dresden.de)

Actin molecules are the central building blocks of the actin cytoskeleton. These 42 kDa globular proteins, comprised of 4 subdomains (SD 1–4) and a central nucleotide binding site, assemble filamentous polymers in an ATP-dependent manner[1,2]. The nucleation, growth, stability, turnover, and three-dimensional organization of actin filaments is tightly regulated by signaling molecules and ABPs[3,4]. These actin-ABP interactions create a dynamic structural scaffold that regulates many cellular processes including transcription, chromatin remodeling, division, adhesion, migration, endocytosis, intracellular trafficking, and contraction in both muscle and nonmuscle cells[5].

The human genome encodes six highly conserved actin isoforms (>93% similarity), which are produced in a time- and tissue-specific manner. They are classified according to their relative isoelectric focusing mobility and enrichment in striated muscle (α-cardiac actin: α-CAA; α-skeletal actin: α-SKA), smooth muscle (α-smooth muscle actin: α-SMA; γ-smooth muscle actin: γ-SMA), and nonmuscle (β-cytoplasmic actin: β-CYA; γ-cytoplasmic actin: γ-CYA) tissues[6]. For the ubiquitously expressed β-CYA and γ-CYA isoforms, which differ by only four N-terminal amino acids, distinct cellular localizations and functions have been reported[7–10]. In the cytoplasm, γ-CYA enriches in submembranous networks and is associated with cell migration, whilst β-CYA is implicated in contractile processes and accordingly localizes to stress fibers and cell-cell contacts[7–9]. β-CYA is additionally recognized as the nuclear actin isoform (reviewed by Viita and Vartiainen[11]) and knockout experiments in mice have demonstrated its essential role in early embryonic development[12]. Isoform specific differences have been attributed to variances in ACTB and ACTG1 nucleotide sequences[13] (encoding β-CYA and γ-CYA, respectively), mRNA trafficking[14], translational dynamics and post-translational modifications[15,16], polymerization properties[17], and biochemical preferences for specific ABPs[9,18].

Heterozygous constitutive mutations in both ACTB and ACTG1 have been associated with BWCFF, a well-defined syndrome with recognizable facial features, developmental disability, neuronal migration defects, hearing loss, ocular colobomas, heart and renal defects, and progressive muscle wasting[19,20]. Our previous case studies demonstrated isoform specific differences amongst BWCFF patients, whereby ACTB variants are linked to severe forms of the disease and ACTG1 mutations are consistently associated with brain malformations[19,21]. In addition to BWCFF, ACTG1 germ-line mutations are also linked to isolated nonsyndromic hearing loss[22], whilst ACTB haploinsufficiency and a low-grade mosaic ACTB hotspot mutation are associated with intellectual disability and Becker's Nevus Syndrome, respectively[23,24]. A single clinical case study where a constitutive disease-causing ACTB variant was not associated with BWCFF was reported by Nunoi and colleagues[25]. In this instance, a patient with a de novo missense mutation in exon 6 of ACTB presented with moderate intellectual disability, abnormal white blood cell counts with recurrent infections, and thrombocytopenia.

Here, we describe six individuals from four unrelated families carrying de novo or co-segregating heterozygous variants in exons 5 and 6 of the ACTB gene. Patients are clinically distinct from those with BWCFF, presenting with mild developmental disability, unspecific minor facial anomalies, microcephaly and thrombocytopenia with platelet anisotropy (variable size including normal and enlarged platelets). As thrombocytopenia is a distinguishing clinical feature of this cohort, we sought to elucidate how variant β-CYA impacts processes underlying thrombopoiesis. We first utilized patient-derived dermal fibroblasts to dissect the effects of these mutations on cell morphology, behavior, cytoskeletal organization and ABP interactions, and subsequently validated these results in patient-derived megakaryocytes (MKs) and thrombocytes. Our results indicate that β-CYA mutations compromise microtubule organization in proplatelets and preplatelets and by this, inhibit the final stages of platelet maturation.

## Results

**Identification and characterization of ACTB-AST patients.** We ascertained a cohort of six patients with four different heterozygous mutations in ACTB (Table 1), who presented with syndromic developmental disorders but could not be definitively diagnosed with a specific Mendelian disorder, including BWCFF (clinical summary in Supplementary Data 1). In contrast to BWCFF, which is associated with missense mutations within exons 2–4, variants in these patients cluster in the 3′ region of ACTB, within exons 5 and 6 (Fig. 1a, magenta). A single missense mutation in exon 6 of ACTB has previously been reported, with the patient displaying moderate intellectual disability, thrombocytopenia, abnormal white blood cell counts and recurrent infections[25] (Fig. 1a, purple). The variants in our cohort include one missense mutation, two small deletions and a single base pair insertion.

In Family A (Fig. 1b), Patient 1 (P1), a four-year-old boy of Eastern European ancestry, presented with mild developmental delay, incomplete cleft lip, heart defect (patent ductus arteriosus and major aortopulmonary collateral arteries), mild microcephaly, leukocytosis with an increased eosinophil count and thrombocytopenia without spontaneous bleeding. Thrombocyte anisotropy with enlarged, immature platelets was reported. His father, patient 2 (P2), also displayed thrombocytopenia without episodes of spontaneous bleeding. Although P2 reported cardiac catheterization in infancy, his medical history was otherwise unremarkable. He attended a mainstream school and has been employed on a regular basis. Whole exome sequencing revealed a likely pathogenic ACTB missense variant in both patients, c.938T>G (Supplementary Data 2). This mutation encodes p.Met313Arg within subdomain 3 (SD3) of β-CYA (Fig. 1b, right, Supplementary Fig. 1).

**Table 1 Summary of ACTB-AST mutations**

| Patient | Genomic position (hg19) | cDNA (NM_001101.3) | Amino acid change |
|---|---|---|---|
| 1, 2 | chr7:g.[5567681A>C] | c.938T>G | p.(Met313Arg) |
| 3[b], 4[a,b] | chr7:g.[5567499_5567515del] | c.992_1008del | p.(Ala331Val_fs*27) |
| 5[a,b,c] | chr7:g.[5567484_5567495del] | c.1012_1023del | p.(Ser338_Ile341del) |
| 6 | chr7:g.[5567406dup] | c.1101dup | p.(Ser368Leu_fs*13) |
| N[d] | chr7:g.[5567417C>T] | c.1090G>A | p.(Glu364Lys) |

[a]Access to primary dermal fibroblasts for cell-based assays and RNA-sequencing analysis
[b]Access to peripheral blood for megakaryocyte differentiation experiments and thrombocyte assessment
[c]Considered to be the most severely affected patient in this ACTB-AST cohort based on severe microcephaly, prominent facial features, and hematological anomalies
[d]Patient described by Nunoi et al.[25] and the only case where mutant β-CYA has been biochemically characterized[38]

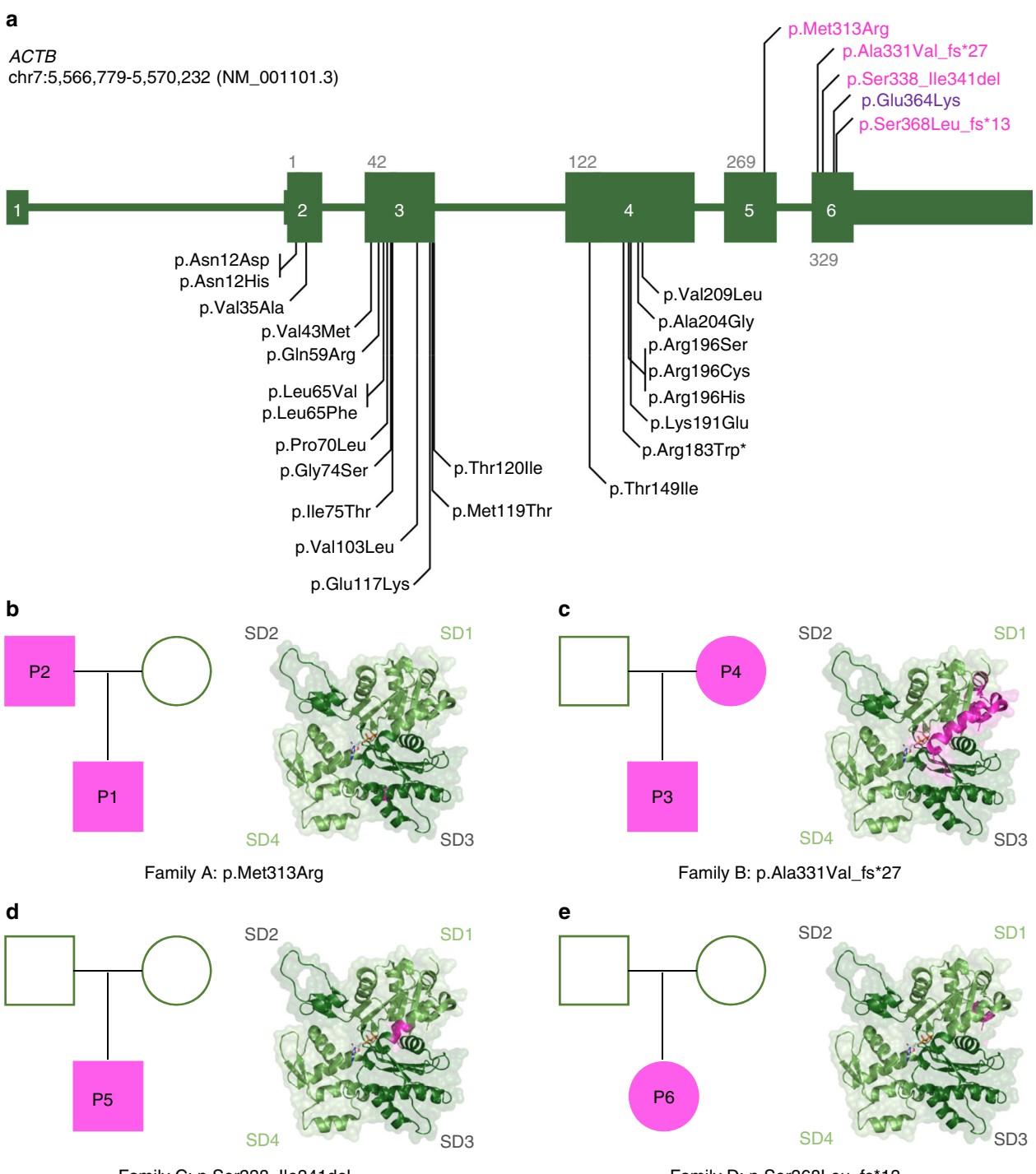

**Fig. 1** Overview of *ACTB*-AST mutations. **a** Schematic representations of *ACTB* mutations. *ACTB*-AST mutations are shown in magenta above the gene model and BWCFF-associated mutations from the literature and from our own patients are given below the gene model; Asterisk (*) indicates a specific variant associated with progressive dystonia[67]. Genomic coordinates refer to the GRCh37/hg19 genome assembly. Exons are numbered and coding exons are indicated by large boxes; **b–e** Pedigree charts (left) and in silico representations of impacted residues (right) in **b** Family A (P1 and P2)—*ACTB*: p. Met313Arg, **c** Family B (P3 and P4) - *ACTB*: p.Ala331Val_fs*27, **d** Family C (P5)—*ACTB*: p.Ser338_Ile341del and **e** Family D (P6)—*ACTB*: p.Ser368Leu_fs*13. For pedigree charts, squares represent males, circles indicate females, magenta-shaded symbols indicate individuals with *ACTB*-AST and patients are numbered according to the text. For in silico representations, affected amino acid residues are indicated in magenta on the actin protein structure (PDB 5JLH)

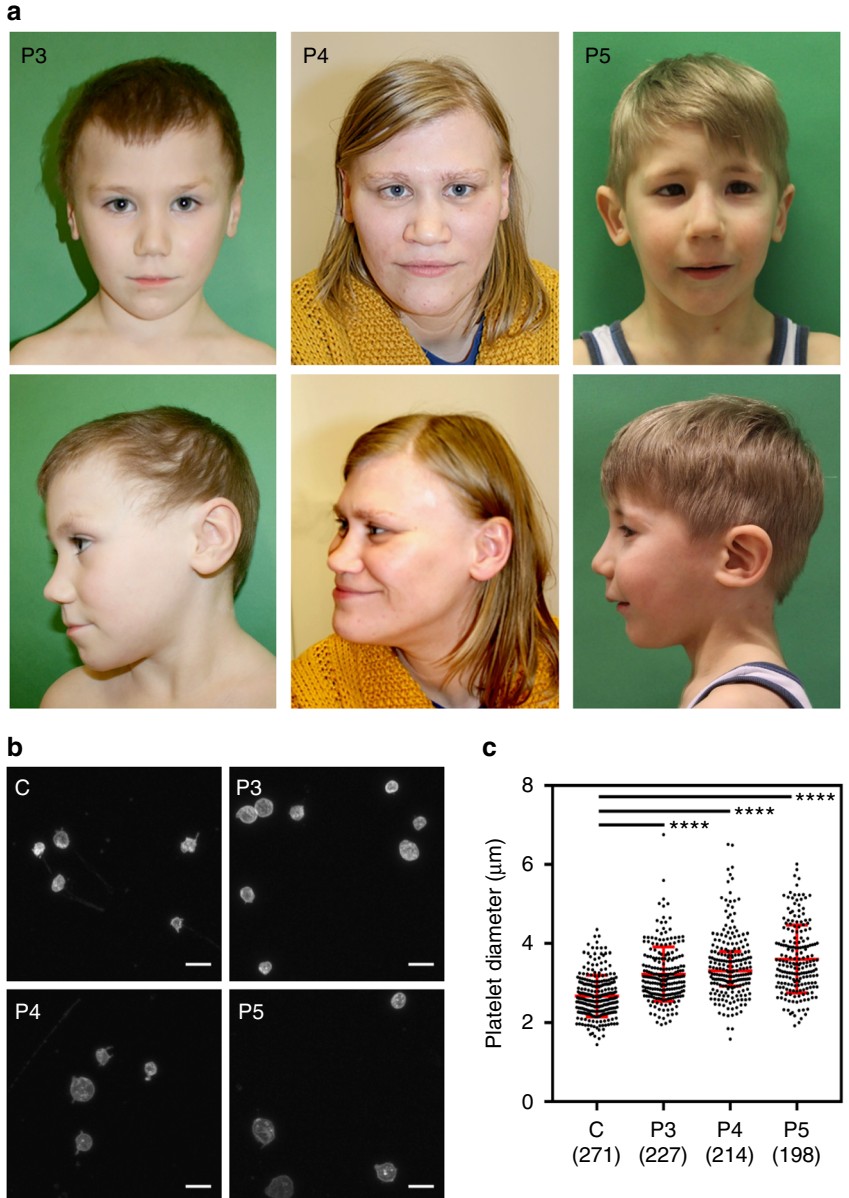

**Fig. 2** *ACTB*-AST patients display minor facial anomalies and thrombocytopenia with enlarged platelets. **a** Craniofacial appearance of patient 3 (left, P3, p. Ala331Val_fs*27) at 5 years of age, patient 4 (mid, P4, p.Ala331Val_fs*27) at 31 years of age and patient 5 (right, P5, p.Ser338_Ile341del) at 4 year 10 months. Flared eyebrows (P3 and P4), straight eyebrows (P4 and P5), telecanthus (all), epicanthal folds (P3 and P5), upslanting palpebral fissures (P3 and P4), a broad nasal tip (P3 and P5), a bulbous nose (P4), a thin upper vermillion border (all) and prominent chin (P4) are observed in these patients; **b**, **c** CD61-labeled platelets purified from a healthy control (C), P3, P4 and P5 were analyzed by immunofluorescence microscopy; **b** Representative images show that platelets in patient samples vary from normal to large in size. Scale bars represent 5 μm; **c** Particle analysis shows a significant shift in the size distribution and average diameter of patient platelets compared with a healthy control. Individual data points are plotted with the median and IQR. The number of platelets analyzed from 1 experiment is given in brackets below each condition. Significance was determined with the Kruskal–Wallis and Dunn's multiple comparisons tests, where ****$p < 0.0001$

In Family B (Fig. 1c), Patient 3 (P3), a five-year-old boy of Central European ancestry, presented with speech delay, borderline intellectual impairment and microcephaly. After an uncomplicated ulnar fracture, he displayed slow bone healing and developed secondary pseudarthrosis. His mother, Patient 4 (P4), also displayed mild intellectual disability (requiring daily support and employment in a sheltered environment), microcephaly and has a history of thrombocytopenia with prolonged postoperative bleeding reported but no spontaneous bleeding events. Both patients display minor facial anomalies, including flared

eyebrows, telecanthus, upslanting palpebral fissures and a thin upper vermillion border (Fig. 2a). Whole exome sequencing demonstrated a likely pathogenic frameshift variant, c.992_1008del, in the last exon of *ACTB* in both patients (Supplementary Data 2). This mutation results in a frameshift leading to the substitution of 26 amino acids from position 331 and the introduction of a premature stop codon leading to truncation at position 357 of β-CYA, within subdomain 1 (SD1) (Fig. 1c right, Supplementary Fig. 1). Thrombocytopenia was tested in P3 after identification of the *ACTB* mutation. Like P4, P3

demonstrated low platelet count with platelet anisotropy, which included large platelets (Fig. 2b).

Patient 5 (P5, Fig. 1d), a five-year-old boy of Central European ancestry, presented with congenital microcephaly that progressed into severe postnatal microcephaly. He showed multiple minor facial anomalies (Fig. 2a; straight eyebrows, telecanthus, bilateral epicantic folds, broad nasal tip and thin upper lip vermilion) and had significant delays in speech development, which progressed with combined speech and physical therapy. This patient also displayed hematological anomalies including leukocytosis with increased eosinophil count, monocytosis, and thrombocytopenia. However, he did not have a history of recurrent infections or spontaneous bleedings. P5 had platelet anisotropy with enlarged platelets (Fig. 2b), an elevated fraction of immature platelets in the peripheral blood (Supplementary Data 1) and bone marrow examination showed increased MK count (Supplementary Fig. 2). Whole exome sequencing revealed a de novo in frame deletion in the last exon of *ACTB*, c. 1012_1023del (Supplementary Data 2), which results in the deletion of resides 338–341 within SD1 of β-CYA (Fig. 1d right, Supplementary Fig. 1).

Patient 6 (P6, Fig. 1e), a five-year-old girl of Western European origin, presented with early developmental delay, microcephaly and a history of recurrent thrombocytopenia during the first year of life, which normalized spontaneously. She made good developmental progress following intensive combined therapies, achieving a low normal IQ at 5 years of age. Brain MRI demonstrated two unilateral periventricular nodular heterotopias with otherwise normal brain morphology. No seizures were documented at the last follow up. Whole exome sequencing revealed a de novo insertion in the last exon of *ACTB*, c.1101dup (Supplementary Data 2), resulting in the substitution of 8 amino acids and addition of 4 residues at the far C-terminus of β-CYA (Fig. 1e right, Supplementary Fig. 1).

Common features amongst this cohort of patients with 3′ *ACTB* variants include developmental delay, mild intellectual disability, microcephaly, and thrombocytopenia with platelet anisotropy and enlarged platelets (Fig. 2b, c). Given the distinct genotype–phenotype correlation, we name this actinopathy *ACTB*-associated syndromic thrombocytopenia (*ACTB*-AST).

**Fibroblasts as a cellular model for *ACTB*-AST.** To assess and model the effects of *ACTB*-AST variants at the cellular level, primary dermal fibroblasts were harvested from P4, P5 and a healthy control. This cell type is robust, can readily be obtained with a minimally invasive skin biopsy and has been utilized to model disease states[26]. At high density, the three control and patient cultures are morphologically indistinguishable. However, the *ACTB*-AST fibroblasts are visibly smaller than healthy control cells at low density (Fig. 3a). Quantitative immunofluorescence microscopy shows that compared to the control, the substrate surface area is reduced by 21.6% and 43.2% for P4 and P5, respectively (Fig. 3b, measured at 50–70% confluency). This is supported by flow cytometry analysis, which shows an ~25% reduction in cell volume for both patient cultures (Fig. 3c, collected at 70% confluency; see Supplementary Fig. 3 for gating strategy). In addition to their small size, P5 fibroblasts form distinct clusters (Fig. 3a, arrow). Live cell migration experiments reveal that whilst control and P4 fibroblasts transiently interact with neighboring cells, P5 fibroblasts cluster due to strong intercellular interactions (Supplementary Fig. 4a). These experiments additionally show that the trajectory, speed and persistence of *ACTB*-AST fibroblasts is significantly altered compared to healthy control fibroblasts (Fig. 3d). No significant differences in cell proliferation rate are observed between the three cultures (Supplementary Fig. 4b).

**Actin isoform expression and compensation in *ACTB*-AST cells.** The effect of *ACTB*-AST mutations on *ACTB* mRNA transcript levels and β-CYA protein expression was assessed in patient fibroblasts. With whole transcriptome sequencing (RNA-Seq, also see Supplementary Note 1 and Supplementary Fig. 5), we confirm that the P4 and P5 variants do not result in nonsense-mediated mRNA decay (Supplementary Fig. 6a). Translation of variant mRNA is demonstrated for P4, as the corresponding C-terminal frame shift peptide was detected in P4 fibroblast lysates by mass spectrometry (Supplementary Fig. 6b). However, the peptide region affected in P5 could not be detected in either control or P5 fibroblast lysates. In patient fibroblasts, *ACTB* mRNA is significantly upregulated, with $3.0 \pm 1.1$ and $2.6 \pm 1.1$ (mean ± s.e.m.) fold changes detected for P4 and P5, respectively (Fig. 4a). Total protein analysis indicates that elevated transcript levels are compensating for reduced β-CYA in patient cells (Fig. 4b), where decreases of 23 and 36% are detected for P4 and P5, respectively (Fig. 4c).

Several studies have demonstrated that actin isoforms exist in equilibrium with one another to ensure that the cells total actin pool is constantly maintained. In the case of β-CYA knockdown and knockout, strong compensatory changes in γ-CYA and α-SMA expression have been reported[7,9,12]. Here, we observe similar effects in *ACTB*-AST patient fibroblasts (Fig. 4a–c). RNA-Seq shows significant increases in *ACTG1* and *ACTA2* mRNA (encoding γ-CYA and α-SMA, respectively) in P4 fibroblasts compared to the control; 1.9 and 1.7 fold changes were recorded (Fig. 4a). Whilst there is a slight but insignificant increase in *ACTG1* mRNA in P5, *ACTA2* is significantly upregulated (fold change = $2.02 \pm 1.1$ s.e.m). With western blot analysis, we show that β-CYA and γ-CYA are the two most abundant actin isoforms in all cell cultures. In patient fibroblast lysates, both γ-CYA and α-SMA protein levels are significantly upregulated, and total actin remains unchanged (Fig. 4b, c). Whilst whole transcriptome sequencing shows a significant upregulation of *ACTG2* mRNA (encoding γ-SMA) in patient samples, this is biased by low transcript levels in the control (raw counts 0–2). Accordingly, γ-SMA is not detected in any sample by either western blot or immunofluorescence analyses. *ACTA1* and *ACTC1* genes, encoding α-SKA and α-CAA, are not expressed in these cells.

In accordance with the western blot results, immunofluorescence analysis also demonstrates a shift in the isoactin equilibrium of *ACTB*-AST fibroblasts, with the β-CYA:γ-CYA ratio decreasing by 30% for both P4 and P5 (Fig. 4d). In agreement with previous reports[8], these isoforms display discrete lateral and axial segregation in both control and patient fibroblasts. Whilst γ-CYA enriches at the cell periphery and beneath the plasma membrane, β-CYA localizes to filament populations at the cells' basal membrane, which include the sub-nuclear filament population (Supplementary Fig. 7a and Fig. 4e). These basal β-CYA filaments are phenotypically consistent between healthy control and BWCFF control fibroblasts (p.Thr120Ile; P1 in[21]). However, in *ACTB*-AST cells they are bundled into thick fibers (Fig. 4e, arrows). Moreover, immunofluorescence assessment of α-SMA shows elevated protein expression in *ACTB*-AST cells (Fig. 4f), increasing by 1.7-fold ± 0.1 in P4 cells and 1.8-fold ± 0.1 in P5 cells (Fig. 4g, mean ± s.d.). This upregulation is not observed in a BWCFF control culture (0.9-fold ± 0.1 s.d.). α-SMA localizes to basal sub-nuclear filaments in ~ 80% of P4 and P5 cells (Fig. 4h). Colocalization analysis shows greater overlap between α-SMA and β-CYA compared to γ-CYA, indicating preferential incorporation of α-SMA into β-CYA bundles in *ACTB*-AST fibroblasts (Fig. 4i, j and Supplementary Fig. 7b).

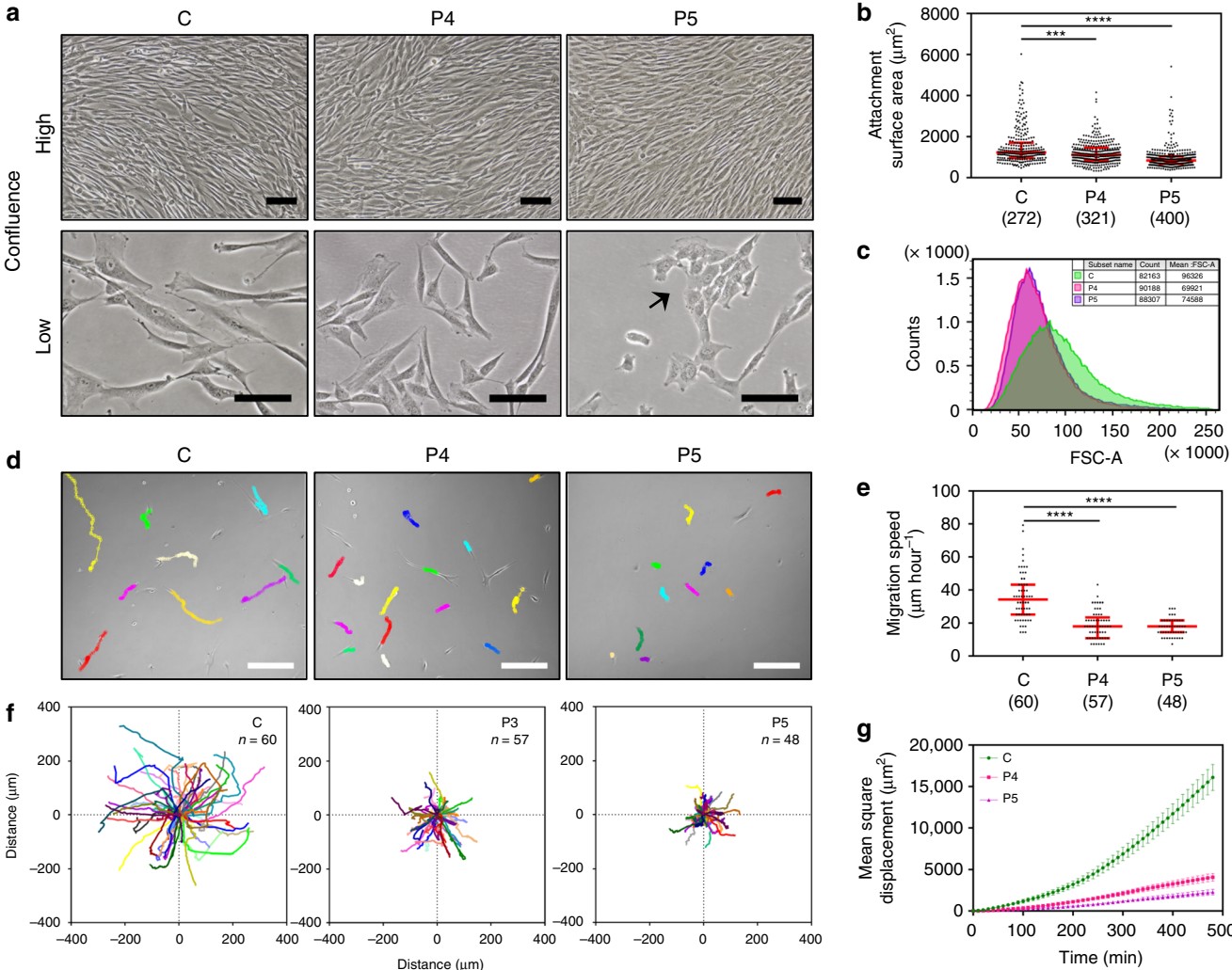

**Fig. 3** Reduced cell attachment surface area, volume and migratory capacity of *ACTB*-AST fibroblasts. **a** Micrographs of control (C), patient 4 (P4, p. Ala331Valfs*27) and patient 5 (P5, p.Ser338_Ile341del) primary dermal fibroblasts at high (top row) and low (bottom row) confluence. At low confluence, *ACTB*-AST cells are distinctly smaller than controls and P5 cells grow in aggregates (arrow). All scale bars are 100 μm; **b** Quantification of the cell attachment surface area from immunofluorescence analyses (i.e. Figure 4e) shows reduced coverage distribution by *ACTB*-AST cells (median and IQR, the number of cells analyzed in 4 experiments is given in brackets); **c** Flow cytometry analysis of the Forward Scatter Area (FSC-A) vs. normalized cell count (100,000 events from 1 experiment) shows a reduction in *ACTB*-AST cell volume (P4: pink, P5: purple) compared to the control (green); **d–g** Migration assays demonstrate reduced migratory capacity for *ACTB*-AST patient primary fibroblasts; **d** Representative images at 8 h with migratory tracks overlaid. Scale bars are 100 μm; **e** Migration speed of individual cells represented in μm per hour (median and IQR); **f** Trajectories of all tracks recorded for C, P4 and P5 from 0 h (origin) to 8 h (5 movies from 2 technical replicates, *n* = number of cells analyzed); **g** Mean square displacement analysis of C (green), P4 (magenta) and P5 (purple) fibroblasts (mean ± s.e.m.). Significance was determined with the Kruskal–Wallis test, where \*\*\**p* < 0.001 and \*\*\*\**p* < 0.0001

**ABP recruitment to affected filament populations**. Whilst the structural scaffold of the actin cytoskeleton is predominantly defined by actins, ABPs govern important aspects of cytoskeletal dynamics and function. ABPs regulate key steps during thrombopoiesis and maintain the structural integrity and functional capacity of circulating thrombocytes (reviewed in the ref. [27–29]). As such, we sought to define a shortlist of ABPs affected by variant β-CYA in P4 and P5 fibroblasts that contribute to the thrombocytopenia phenotype observed in *ACTB*-AST patients. We approached this task by assembling a list of deregulated ABP-encoding RNAs (based on those listed by Winder et al.[3]), which were cross-checked by their clinical association with thrombocytopenia and validated at the protein level in cellular assays. Amongst the shortlist of 50 deregulated genes, five candidates were identified, which have mutations associated with

thrombocytopenia and enlarged platelets (Fig. 5a, red boxes). These genes encode for α-actinin 1[30,31], nonmuscle myosin-2A (NM-2A)[32,33], diaphanous formin-1 (Diaph1)[34], filamin A[35,36], and tropomyosin (Tpm)4.2[37]. The upregulation of four of these candidate genes at the protein level in P4 and P5 cells is demonstrated with western blot analysis (Fig. 5b, c). Immunofluorescence assessment of these four proteins shows the enrichment of NM-2A, and recruitment of both α-actinin 1 and filamin A, into basal β-CYA/α-SMA bundles in both P4 and P5 fibroblasts (Fig. 5d). Conversely, an antibody recognizing Tpm4.1/4.2 isoforms does not label these structures. Quantification of ABP fluorescence within the sub-nuclear region of individual cells supports these observations (Fig. 5e). Specifically, significant increases in α-actinin 1, NM-2A and filamin A fluorescence are observed in P4 and P5, whilst the level of

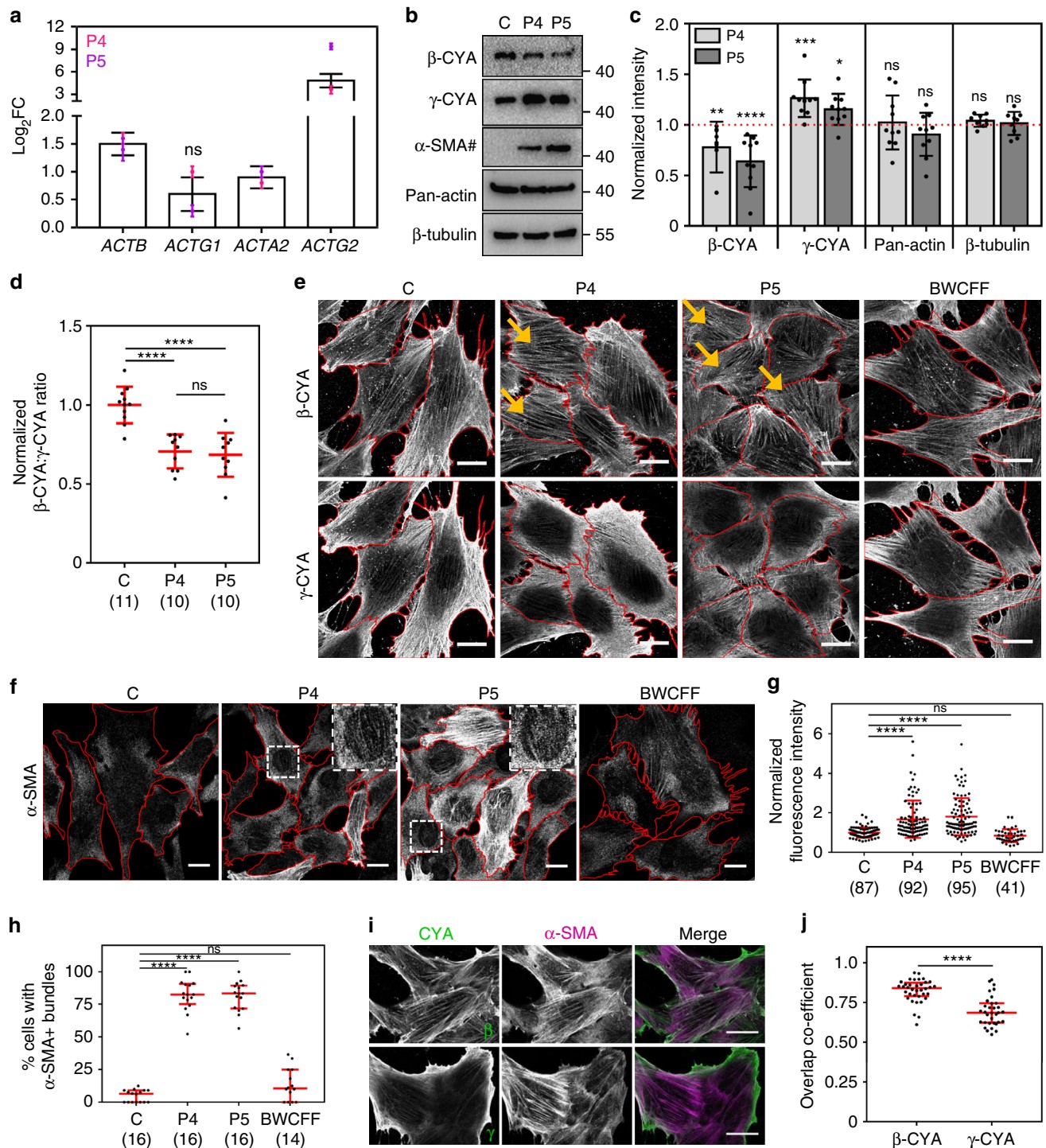

Tpm4.1/4.2 is unchanged in P5 and even downregulated in P4. This sub-nuclear phenotype observed in P4 and P5 cells is distinct from that visualized in the BWCFF control.

**Molecular modeling of P4 and P5 β-CYA and affected ABPs.** In silico approaches were used to elucidate the molecular basis underlying our cellular findings. Our previous study, which biochemically characterized the disease-associated *ACTB* p.Glu364-Lys and p.Arg183Trp mutations, demonstrated that these modeling approaches are in good agreement with experimental results[38]. Here, predictions of the mutation-mediated structural changes in β-CYA from P4 and P5 (β-CYA[P4] and β-CYA[P5])

show that structural changes are localized in SD1, a region that critically contributes to actin-ABP interactions (Fig. 6a and Supplementary Note 1). Specifically, NM-2 and α-actinin/filamin, whose expression and localization were shown to be altered in cellular studies, bind to the affected SD1 region (Fig. 6b, c). In the case of NM-2, our model suggests significant changes in the interaction of β-CYA[P4] and β-CYA[P5] with the myosin cardio-myopathy (CM)-loop and supporting-loop[39,40] (Fig. 6b and Supplementary Fig. 8). In the case of β-CYA[P4], the altered structure retains critical features of the interaction with the supporting-loop (see Supplementary Note 1). For α-actinin/fila-min A, a critical hydrophobic interaction is conserved in both

**Fig. 4** Actin isoform regulation in *ACTB*-AST patient fibroblasts. **a** RNA-Seq analysis of actin transcripts in P4 and P5 compared to the healthy control ($\log_2$ fold change ± s.e.m.). Bar graphs show the combined results from both patients relative to the control. Overlayed dot plots show the individual patient data calculated from three technical replicates ($p < 0.0001$ unless stated as ns); **b** Representative western blots show reduced β-CYA, increased γ-CYA and α-SMA in P4 and P5 fibroblasts compared to the control (C). Double the sample amount was required to detect α-SMA (#); **c** Densitometry analysis of P4 and P5 signals expressed as mean (±s.d.) relative to the control (9–10 replicates from 6 lysates; Kruskal–Wallis test); **d** Reduced β-CYA:γ-CYA ratios are detected in P4 and P5 fibroblasts by immunofluorescence microscopy (mean ± s.d.; image numbers analyzed from 3 experiments are given in brackets; one-way ANOVA); **e** Representative maximum intensity projections show β-CYA (top) and γ-CYA (bottom) distribution within z-stack slices 1–2 of C, P4, P5, and BWCFF control fibroblasts. Yellow arrows indicate cells where thick basal β-CYA bundles are abundant. Scale bars are 15 μm; **f** Immunofluorescence microscopy shows increased α-SMA expression in P4 and P5 fibroblasts. Inserts show α-SMA incorporation in *ACTB*-AST sub-nuclear basal filaments. Scale bars are 15 μm; **g** Normalized α-SMA fluorescence intensity in individual cells (mean ± s.d.; the number of cells analyzed from 3 experiments is given in brackets; Kruskal–Wallis test); **h** The percentage of cells per image in which α-SMA is incorporated into basal sub-nuclear filaments (median and IQR; 14–16 images from 3 experiments; Kruskal-Wallis test); **i** P5 cells co-stained for β-CYA or γ-CYA (left, green in merge) and α-SMA (mid, magenta in merge). Scale bars are 20 μm; **j** Greater overlap of α-SMA with β-CYA is observed in P5 cells (median and IQR; 33–36 cells from 3 experiments; Mann–Whitney test). In all cases, *$p < 0.05$, ***$p < 0.001$, ****$p < 0.0001$ and ns, not significant

β-CYA$^{P4}$ and β-CYA$^{P5}$. In the case of β-CYA$^{P4}$, the local rearrangements bring two aspartate residues in hydrogen-bonding distance to surface residues on α-actinin (Fig. 6c and Supplementary Note 1).

**Characterization of *ACTB*-AST platelets**. With this cytoskeletal foundation delineated in fibroblasts, we sought to validate our results in disease-relevant thrombocytes. As such, thrombocyte cytoskeletal constituents and ultrastructure were assessed in platelets purified from control, P3, P4 and P5 peripheral blood (Fig. 7, Supplementary Fig. 9, Supplementary Fig. 10 and Supplementary Note 1). In accordance with previous reports, we observe actin throughout resting control platelets[41], with both β-CYA and γ-CYA found at the cortex and within the platelet core (Fig. 7a). Patient platelets, which are frequently enlarged (see Fig. 2c for size distribution), have less β-CYA levels compared to controls. Quantitative analysis of projected images indicates that the β-CYA:γ-CYA ratio decreases by 32–37% in patient cells (Supplementary Fig. 9a). Clear cortical redistribution of both isoforms is seen within the mid-plane of patient platelets (Fig. 7a, bottom row). Whilst α-SMA was upregulated and recruited to β-CYA filaments in patient primary dermal fibroblasts, no changes were observed in purified thrombocytes (Supplementary Fig. 9b). Despite this finding, candidate ABP localization patterns are consistent with the fibroblastic studies. Specifically, α-actinin 1, NM-2A, and to a lesser extent filamin A, are all recruited to cortical β-CYA-rich filament populations in patient platelets (Fig. 7b). Tpm4.1/4.2, which does not localize to sub-nuclear β-CYA filaments in patient fibroblasts, is also not recruited to cortical filaments in patient platelets (Supplementary Fig. 9c).

A fine balance between actin and microtubule cytoskeletal forces is pivotal for regulating platelet size during maturation and following activation[41,42]. In healthy resting platelets, a band of microtubules localizes at the platelets cortex, maintaining its discoid shape[43]. This cortical band is perturbed in diseases with giant platelets, where it is significantly thickened in the case of Gray platelet syndrome and disordered like a "ball of yarn" in Epstein's syndrome and May-Hegglin anomaly (caused by NM-2A mutations) platelets[44]. As *ACTB*-AST platelets include those of enlarged size, we assessed β-tubulin localization and organization in control and patient cells. Our analysis shows microtubules in the typical cortical band in control platelets, but in a multitude of different organization patterns in patient platelets, all of which are highly disordered (Fig. 7c).

**ACTB*-AST patient-derived megakaryocytic cells**. During thrombopoiesis, the actin cytoskeleton is critical for MK differentiation and migration to the vascular niche, endothelial barrier penetration, proplatelet branching and tip formation, as well as the final stages of platelet maturation from circulating proplatelet and preplatelet precursors[27,28]. To evaluate the effect of *ACTB*-AST variants in this process, MKs were differentiated for 14 days from peripheral blood mononuclear cells (PBMCs) obtained from two healthy controls, P3, P4, and P5 ($n = 1$). Cells with increased size, typical of polyploid MKs, are apparent in all cultures from day 10 of the differentiation period (Supplementary Fig. 11a). At day 14, CD41$^+$CD42a$^+$CD61$^+$ MK yields in the range of 15–60% were achieved for the 5 cultures (Supplementary Fig. 11b-c).

Consistent with our fibroblast and thrombocyte findings, cells differentiated from patient samples show decreased β-CYA:γ-CYA ratios (Fig. 8a). Reductions of 32% ± 1.4, 25% ± 0.5 and 30% ± 0.8 (mean ± s.d.) are observed for P3, P4, and P5, respectively (Supplementary Fig, 12a). In podosome-forming cells from controls, β-CYA is localized diffusely throughout the cytosol and enriches in basement-membrane degrading podosomes (Supplementary Fig, 12b). Interestingly, in patient cells, these podosomes remain intact and enriched for β-CYA, whilst the cytosolic β-CYA pool is depleted. In proplatelet-forming MKs from controls, both β-CYA and γ-CYA are found within the cell body and throughout proplatelet shafts, with β-CYA enriched in proplatelet swellings (Fig. 8a). In the case of patient cells, proplatelet structures incorporate comparatively less β-CYA than controls and frequently display irregularly shaped swellings (Fig. 8a, bottom row, white arrows). Whilst compensatory α-SMA upregulation is observed in the fibroblastic model cultures (Fig. 4f, g), no change in α-SMA expression is evident in patient-derived MK (Supplementary Fig. 12c). Overall, no remarkable differences in proplatelet number, length or bifurcation frequency are evident between control and patient samples.

Microtubules localize along the entire length of proplatelets, within the shafts, swellings and at the tips, and form coiled structures consistent with the marginal band seen in shed platelets[45]. As microtubule organization is compromised in purified platelets, we assessed its organization in proplatelet-producing MKs (Fig. 8b, c). We observed that proplatelet swellings (ranging in size from 3 to 8 μm) have three morphologically distinct microtubule phenotypes: (1) those with a thin cortical microtubule band, (2) those with a thick cortical microtubule band, and (3) those where microtubules are disorganized (Fig. 8b). Each of these phenotypes is observed along the length of the proplatelet and at its tip. Whilst quantification shows no change in the percentage of phenotype 1 structures between control and patient MKs, significant differences are observed between phenotypes 2 and 3 in (Fig. 8c). In the controls, swellings with thick cortical microtubule bands comprise 41% ± 12 (mean ± s.d.) of the events analyzed, whilst disordered structures contribute to less than 10%. Conversely,

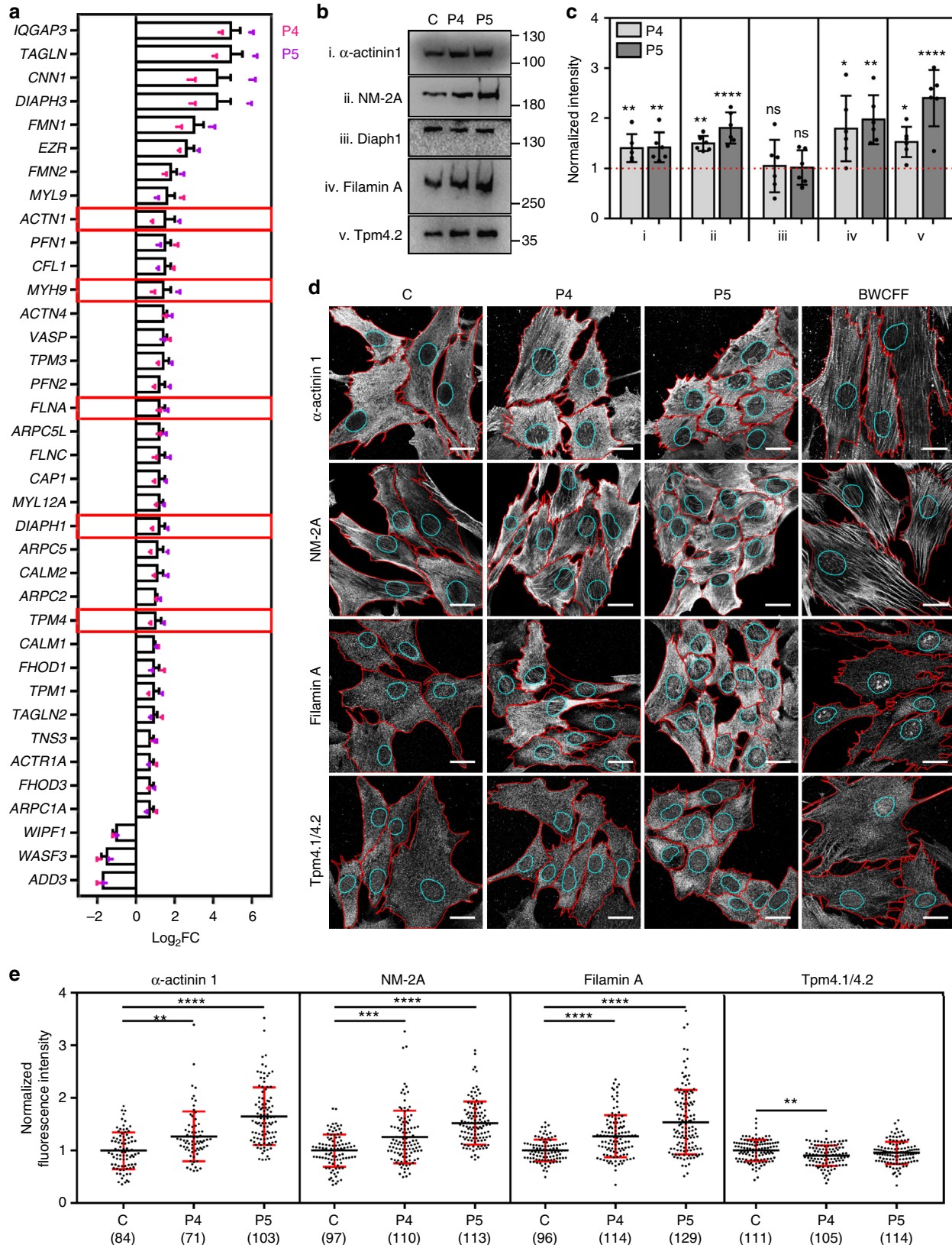

swellings with thick cortical microtubule bands make up only 13–19% of all events in patient MKs, whereas swellings with disordered microtubules are observed at significantly elevated frequencies of 40% ± 12 for P3, 34% ± 9 for P4 and 46% ± 16 for P5 (mean ± s.d. for all).

## Discussion

BWCFF was the first human disease associated with *ACTB* mutations[20]. To date, a broader spectrum of distinct clinical entities associated with *ACTB* have been described. These include intellectual disability due to *ACTB* haploinsufficiency[23], Becker's

**Fig. 5** Select thrombocytopenia-associated ABPs are recruited to basal β-CYA-rich filaments. **a** ABP transcripts significantly deregulated (log$_2$ fold change ± s.e.m.) in both P4 and P5 fibroblasts compared to a healthy control, as determined by RNA-Seq. Bar graphs show the combined results from both patients relative to the control. Dot plots show the individual patient data calculated from three technical replicates. Red boxes indicate the genes where disease-causing mutations have been associated with thrombocytopenia with enlarged platelets; **b**, **c** Western blot analysis and densitometry of ABP candidates: (i) α-actinin 1, (ii) NM-2A, (iii) Diaph1, (iv) Filamin A and (v) Tpm4.1/4.2 in control (C), P4 and P5 fibroblast lysates. Significant upregulation is validated for all candidates except Diaph1. Data are represented relative to the control (mean ± s.d.; 6 replicates from 3 lysates; Kruskal–Wallis test); **d** Representative maximum intensity projections show α-actinin 1 (top row), NM-2A (second row), Filamin A (third row) and Tpm4.1/4.2 distribution within z-stack slices 1–2 of C, P4, P5 and BWCFF control fibroblasts. Cell boundaries are shown in red and cyan regions indicate the nuclear boundaries where basal sub-nuclear filaments localize. Scale bars are 20 μm; **e** Quantification of the fluorescence intensity of each candidate ABP in the sub-nuclear region (mean ± s.d.; Kruskal–Wallis test). The number of cells analyzed from 3 experiments is indicated in brackets. In all cases, **p < 0.01, ***p < 0.001 and ****p < 0.0001

Nevus syndrome due to low-grade mosaic *ACTB* hotspot mutations[24], BWCFF resulting from constitutive missense mutations in exons 2–4[19–21], and a single case where moderate intellectual disability, white blood cell anomalies and thrombocytopenia are linked to a missense mutation in exon 6[25]. Due to clinical consistencies between this patient and our cohort (Supplementary Data 1, Patient N), and the fact that the exon 6 variant identified falls within our 3′ mutation cluster, we propose that this patient was the first reported case of *ACTB*-AST. The gnomAD database lists 7 additional heterozygous variants in the 3′ region of *ACTB* (Supplementary Table 1), each of which is seen in a single individual. It is not currently known whether these variants are truly benign, not fully penetrant, or causative for *ACTB*-AST (Supplementary Note 1).

We define *ACTB*-AST as a clinical entity characterized by mutations in the 3′ region of *ACTB*, within exons 5 and 6. Patients' symptoms include microcephaly, intellectual disability, minor facial anomalies, white blood cell anomalies and thrombocytopenia. Symptoms and their severity vary greatly between affected patients, even between patients with the same mutation. Most patients presented with mild microcephaly (P1, P3, P4, and P6), with a severe form observed only in P5. Intellectual disability is also mild to borderline and P2 from family A also showed normal intellectual function. For P5 and P6, where developmental milestones were significantly delayed, good developmental progress was observed following intensive therapeutic intervention. The minor facial anomalies observed in these patients show no overlap with the striking and recognizable facial features associated with BWCFF. Our observation is that the facial phenotype is not sufficiently specific to clinically diagnose *ACTB*-AST. Furthermore, thrombocytopenia in these patients was only revealed during routine checkup, was never associated with episodes of spontaneous bleeding and in the case of P6, resolved itself during early childhood. For P5, we demonstrate that thrombocytopenia is likely alleviated by increased megakaryopoiesis. However, this remains to be validated for other *ACTB*-AST patients.

For *ACTB*-AST variants, p.Ala331Val_fs*27 (P3 and P4) and p.Ser338_Ile341del (P5), our combined results bridge clinical evaluation with cellular and molecular characterizations (summarized in Supplementary Data 3). Thus far, β-CYA disease-associated mutations have been biochemically characterized in a single study by our group[38]. These variants include the p. Glu364Lys mutant (Patient N) described by Nunoi et al, which is now incorporated into the *ACTB*-AST cohort[25]. We demonstrated that β-CYA$^{PN}$ can be produced in the baculovirus/*Sf*9 expression system, showed that its folding is minimally affected, and verified its functional competence; i.e. it incorporates into filaments and supports myosin based contractility[38]. With mass spectrometry, we were able to validate the production of β-CYA$^{P4}$ in patient dermal fibroblasts. Molecular dynamics simulations predict that β-CYA$^{P5}$, which has a smaller structural perturbation than β-CYA$^{P4}$, is properly folded (see in silico analysis in

Supplementary Note 1). In the Nunoi et al. study, β-CYA$^{PN}$ was shown to be produced along with β-CYA and γ-CYA in patient fibroblasts, platelets and leukocytes. Mutant β-CYA was estimated to be produced at levels equal to that of the β-CYA$^{WT}$[25]. Moreover, our cellular and in silico ABP data provide indirect evidence that mutant β-CYA$^{P4}$ and β-CYA$^{P5}$ are incorporated into actin filaments. Thrombocytopenia-associated filamentous-actin binding proteins identified in cellular expression and localization studies each interact with actin at the interface affected by the mutations.

Primary dermal fibroblasts proved to be a robust and valuable model as they allowed us to: (1) assess phenomena that are difficult to examine in disease-relevant cell types (i.e., growth rate and migratory capacity), (2) screen and predefine variant-affected cytoskeletal filament populations, and (3) statistically test observations prior to validation in limited-access patient material. We observed that fibroblasts display similar β-CYA/γ-CYA equilibrium relationships and β-CYA-ABP interactions as observed in MKs and thrombocytes. Whilst α-SMA compensation in fibroblasts proved to be tissue-specific and not relevant to thrombopoiesis, it may somewhat contribute to the increased contractile activity, enhanced cell-cell contacts (P5) and reduced migratory capacity (P4 and P5) seen in patient fibroblasts[12,46,47]. The fact that proliferation is normal in fibroblasts suggests that it is likely not a major contributor to disease phenotype expression. In contrast, the cells' inability to polarize and move persistently appears to be more critical. Furthermore, the observation that fibroblasts show clear deficits at low confluence and not at high confluence shows how the mutations affect mechanotransduction. Polarization, movement and morphology are greatly affected at the single cell level, whilst the wild-type phenotype is rescued with cell–cell contact formation. Overall, our fibroblast experiments uncovered a membrane-associated cytoskeletal filament population uniquely affected in *ACTB*-AST cells, comprising β-CYA and thrombocytopenia-associated ABPs, such as α-actinin 1, NM-2A and filamin A.

Previous studies have linked mutations in each of the affected ABPs with defects in proplatelet tip formations[30,35,36,48,49]. Reduced bending and bifurcation of the proplatelet shaft results in the release of fewer, larger thrombocytes, and in the case of NM-2A mutations, immature fractions are elevated[30,36,48–52]. Based on these reports, we expected platelet production to be associated with reduced proplatelet tip formation in *ACTB*-AST MKs. However, this hypothesis is not supported by our findings, as mutations show no effect on proplatelet number or branching. Further, we could not link thrombocytopenia in *ACTB*-AST patient cells to differentiation or podosome defects. As our results show reduced migratory capacity for patient fibroblasts and Nunoi et al. reported reduced neutrophil chemotaxis[25], it is reasonable to hypothesize that *ACTB*-AST mutations affect MK migration.

The work of Italiano and colleagues demonstrates that platelets mature from two interchangeable intermediates in circulation:

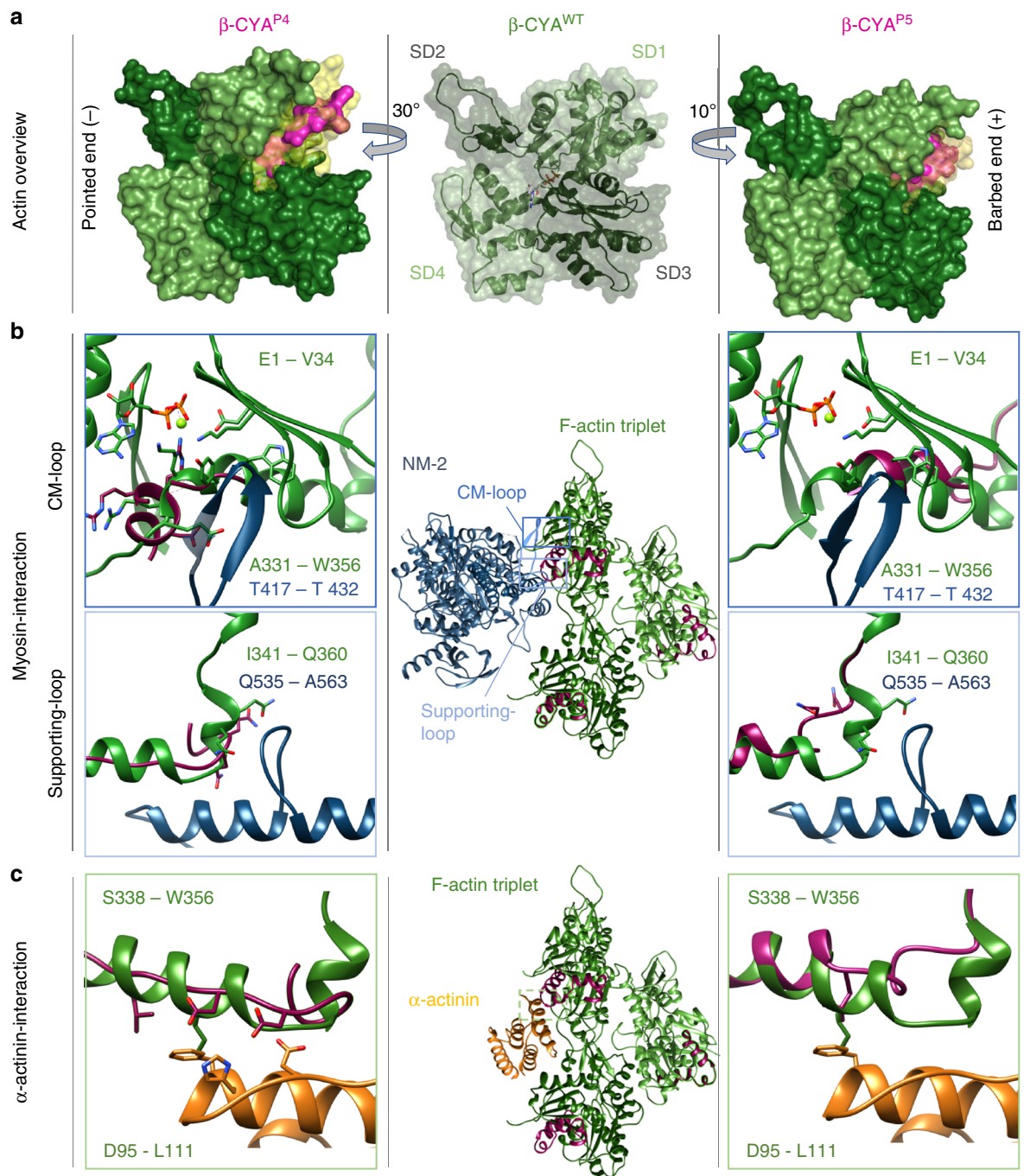

**Fig. 6** In silico modeling of P4- and P5-associated *ACTB* variants. **a** Overview of wildtype (WT) β-CYA (mid) in cartoon with space-fill overlay. β-CYA^P4 and β-CYA^P5 to the left and right respectively, with missing volumes indicated in yellow space-fill and affected residues modeled in magenta space-fill; **b** The binding interface of NM-2 (blue, according to PDB 5JLH) on a β-CYA^WT F-actin triplet (mid, green with magenta C-terminal residues) is shown. Close-ups of affected β-CYA^P4 (left) and β-CYA^P5 (right) residues are superimposed as magenta cartoon structures on the green β-CYA^WT cartoon structure. CM-loop interactions with mutant actins are shown in the top panels, whilst supporting-loop interactions are modeled in the lower panels; **c** In silico modeling of the interaction interface of β-CYA^WT (green) with α-actinin (orange, according to PDB 3LUE, confirmed by docking) is shown (mid). Left and right show the interface with affected C-terminal residues for β-CYA^P4 (left) and β-CYA^P5 (right), respectively

discoid-shaped preplatelets (3–10 μm diameter) and barbell-shaped proplatelets (10–32 μm perimeter)[53]. These are shed following cytoplasmic bridge fission between proplatelet swellings. For preplatelets to convert to barbell-shaped platelets, the cortical microtubule band must bend, twist and be bundled centrally (Fig. 9, top row). This conversion is dependent on microtubule band thickness. Mathematical modeling predicts that this process requires a balanced input of elastic bending forces, microtubule

bundling forces, and actin-myosin-spectrin-mediated membrane tension[42]. In *ACTB*-AST MKs, where β-CYA is depleted from swellings, we observed an elevated fraction of swelling events with disorganized microtubules. Our observations in patient thrombocytes are in line with analyses of giant platelets in patients with May Hegglin anomaly (NM-2A mutation), where microtubules have a loosely packed organization[43]. Based on our observations and the work of Italiano and colleagues[42,53], we propose that

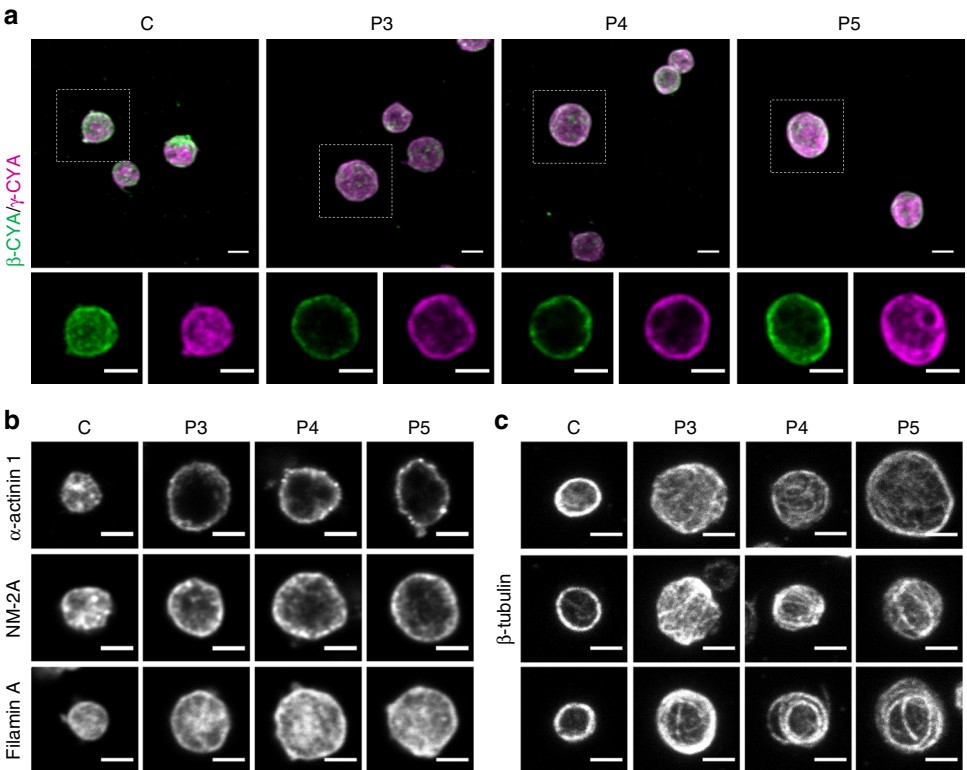

**Fig. 7** *ACTB*-AST patient platelets have disordered actin and microtubule cytoskeletons. **a–c** Assessment of the actin and microtubule cytoskeletons in platelets purified from healthy control (C), patient 3 (P3), patient 4 (P4), and patient 5 (P5) EDTA-peripheral blood (1 experimental replicate); **a** Representative maximum intensity projections (top row) are given for β-CYA (green) and γ-CYA (magenta) labeled samples, showing reduced β-CYA:γ-CYA ratios in patient samples. Mid-stack slices of the marked regions of interest (white squares) are shown below, demonstrating cortical redistribution of actin isoforms in patient thrombocytes; **b** Mid-stack slices show strong cortical recruitment of NM-2A and α-actinin 1, and a moderate cortical enrichment of Filamin A in patient platelets; **c** Representative images demonstrate the highly disordered nature of β-tubulin in patient-derived platelets compared to those from a healthy control. 3–6 images (total of 20–60 platelets) were assessed for each protein of interest. All scale bars represent 2 μm

platelet anisotropy in *ACTB*-AST results from obstruction of final platelet processing steps (Fig. 9, bottom row). Specifically, our data suggest that preplatelets are constrained at larger sizes as microtubules are assembled into unproductive configurations that cannot facilitate barbell-shaped platelet twisting.

In conclusion, this work describes a new clinical phenotype associated with mutations clustered in the 3′ region of *ACTB*, named *ACTB*-AST. Our cross disciplinary approach assesses the impact of pathogenic variants across multiple levels—from the patient, to the cell and then to the molecule. This method allowed us to identify a cytoskeletal phenotype unique to *ACTB*-AST, whereby a specific subset of thrombocytopenia-associated ABPs is recruited to mutant β-CYA bundles. Our data demonstrate that this filament population contributes to thrombopoiesis defects by perturbing microtubule organization in MKs and thrombocytes.

## Methods

**Patient recruitment and whole exome sequencing**. This study was approved by TU Dresden's institutional review board (EK127032017). All procedures were conducted in accordance with the approved guidelines and informed consent of all human research participants. The authors affirm that human research participants provided informed consent for publication of the images in Fig. 2a. Patients presenting with syndromic developmental delay in combination with microcephaly and thrombocytopenia underwent genetic testing in four diagnostic laboratories in Germany and the USA. Probands were not clinically diagnosed with BWCFF after identification of likely pathogenic variants in *ACTB*. Cases were discussed with N. DD regarding the clinical interpretation of *ACTB* variants. P1 and P3 had one parent, father (P2) and mother (P4), respectively, presenting with similar or overlapping clinical features. P5 and P6 were the only affected members of their families. No pathogenic copy number variants were detected with chromosomal microarrays.

All patients received trio-based whole exome sequencing. Exome capture was performed using IDT Xgene exome research panel (P1–P5) and Agilent SureSelectXT Human V5 kit (P6). 150 nt paired-end sequencing was performed with a median target coverage of at least 50-fold on Illumina NextSeq500 Sequencing systems. Alignment (mapping to GRCh37/hg19), variant identification (SNPs and indels), variant annotation and filtering was performed using the CLC Biomedical Genomics Workbench (Qiagen, Hilden, Germany)[54] for patients P1–5 and NextGENe software (SoftGenetics, LLC, State College, PA) with further analysis using Cartagenia Bench Lab NGS software (Agilent Technologies) for P6.

Approximately 65,000 genetic variants were identified per individual. Variants were filtered with a focus on protein-altering variants (missense, frame-shift, splice-site and premature stop-codons) absent from public databases (gnomAD and 1000 Genomes project). In families A (P1 and P2) and B (P3 and P4), 18 and 11 rare familial variants were identified, respectively (Supplementary Data 2). Of these, only *ACTB* variants were considered to be likely pathogenic, according to ACMG criteria[55]. All *ACTB* variants were absent from public databases (see also Supplementary Note 1 on 3′*ACTB* variants listed in the gnomAD database). P5 and P6 had 2 and 3 de novo rare variants, respectively. All *ACTB* variants were confirmed by Sanger sequencing.

**Primary fibroblast cell culture**. Primary dermal fibroblasts were obtained from P4, P5 and a healthy control following 3 mm cutaneous punch biopsies and cultured in BIO-AMF™-2 Medium (Biological Industries USA, Cromwell, CT, USA). For sub-culturing, primary fibroblasts were washed twice with 1× dPBS and detached at 37 °C for 30 s with 0.05% Trypsin/EDTA (Gibco®; Thermo Scientific, Waltham, MA, US). The reaction was stopped with DMEM/F-12 medium (Gibco®; Thermo Scientific) containing 10% FCS (Gibco®; Thermo Scientific) and cells were pelleted by centrifugation at 500×*g* for 5 min. Cells were resuspended in BIO-AMF™-2 medium, seeded onto Corning plasticware (Corning, NY, US) and maintained in BIO-AMF™-2 medium at 37 °C in the presence of 5% CO$_2$. Cultures were continued for a maximum of 5 passages. For the growth curve assay, 3 × 10$^4$ cells were seeded in 24-well plates and counted after 24, 48, and 72 h following trypsinization. Culture images were obtained with a Nikon Eclipse TS100 microscope using 10×/0.25NA and 20×/0.40 NA objectives (Nikon, Minato, Tokyo, Japan). Mycoplasma contamination was routinely tested for by PCR using the

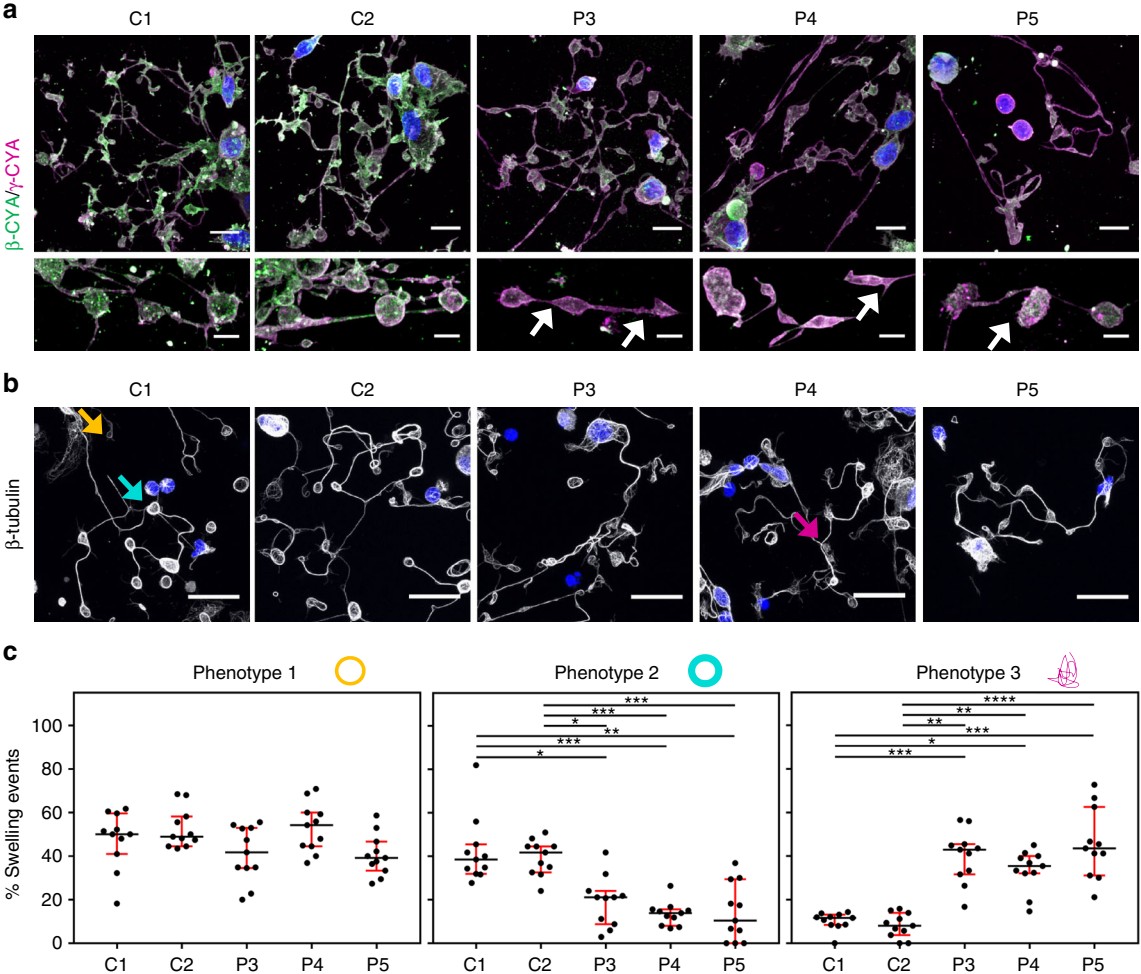

**Fig. 8** Microtubule organization in proplatelet swellings is compromised in *ACTB*-AST-derived megakaryocytes. **a** β-CYA (green) and γ-CYA (magenta) distribution in proplatelet-forming MKs from PBMCs isolated from the whole blood of two healthy controls (C1 and C2), patient 3 (P3), patient 4 (P4), and patient 5 (P5). Representative maximum intensity projections show reduced incorporation of β-CYA into proplatelet structures in *ACTB*-AST cells. White arrows indicate irregular-shaped proplatelet swellings identified in patient samples. Scale bars are 10 μm for the top row and 5 μm for the bottom row; **b** Representative images of β-tubulin labeled MKs from C1, C2, P3, P4, and P5. Arrows indicate the three proplatelet swelling phenotypes observed, where microtubules are (1) organized into a thin marginal band (yellow arrow), (2) organized into a thick marginal band (cyan arrow), or (3) disordered (magenta arrow). Scale bars are 20 μm; **c** Quantification of the percentage of proplatelet-attached swellings with thin marginal bands (phenotype 1, left), thick marginal bands (phenotype 2, middle) and disordered microtubules (phenotype 3, right) shows significant differences between healthy controls and *ACTB*-AST patients. For each condition, blinded analysis of 11 images from 3 samples obtained from 1 experimental repeat was performed. Data are represented as median (IQR). Significance was determined with the Kruskal–Wallis test, where $*p < 0.05$, $**p < 0.01$, $***p < 0.001$, $****p < 0.0001$

Venor®Gem OneStep Kit (Minerva Biolabs GmbH, Berlin, Germany) and with DAPI counterstaining.

**Flow cytometry of fibroblasts**. Patient-derived fibroblasts grown to ~70% confluency were passaged with Accutase™ (StemCell Technologies, Vancouver, Canada). Pelleted and washed cells were fixed with precooled 70% Ethanol and stored at −20 °C. Prior to analysis, cells were labeled with Propidium Iodide (Thermo Scientific). Samples were analyzed on a LSR Fortessa™ (BD Biosciences) of the Flow Cytometry Core Facility, a core facility of BIOTEC/CRTD at Technische Universität Dresden. Data were manually analyzed using FlowJo analysis software (TreeStar, Ashland, OR, USA).

**Fibroblast migration assay**. CELLview™ 4 compartment glass bottom dishes (VWR International, Radnor, PA, US) were pre-coated with fibronectin/gelatin for 1 h at 37 °C. Primary fibroblasts were seeded at a density of $2.5 \times 10^3$ and allowed to attach overnight at 37 °C in the presence of 5% $CO_2$. Cells received fresh BIO-AMF™-2 medium supplemented with 10 mM HEPES and dishes were mounted onto an inverted Nikon Eclipse Ti-E microscope system equipped with a 37 °C incubator (Nikon). Phase contrast images were acquired at 10 min intervals for 8 h with a 10×/0.30NA objective. For analysis, cells were manually tracked with the MTrackJ plugin in ImageJ. Cells were excluded from the analysis if they collided

with another cell or underwent division during the imaging period. The Dicty Tracking 1.4 software package was used to calculate cell trajectories, random migration speeds, directionality ratios and the mean square displacement of cells[56].

**Whole-transcriptome sequencing (RNA-Seq)**. For RNA-Seq, fibroblasts were seeded in 75 $cm^2$ flasks and harvested at ~70% confluence. RNA was extracted using the miRNeasy Mini Kit (Qiagen) according to the manufacturer's instructions. On column DNA digestion was included to remove residual contaminating genomic DNA. All experiments were performed in triplicate, meaning that RNA was independently extracted three times from each culture. For library preparations, the TruSeq Stranded mRNA Library Prep Kit (Illumina) was used according to the manufacturer's protocol, starting with 1 μg total RNA. All barcoded libraries were pooled and sequenced 2 × 75 bp paired-end on an Illumina NextSeq500 platform to obtain a minimum of 10 million reads per sample. Raw reads from Illumina sequencers were converted from bcl to fastq format using bcl2fastq (version v2.17.1.14) allowing for 1 barcode mismatch. Reads were trimmed for quality and sequence adapters using trimmomatic[57]. Trimming resulted in an average of 13.1–22.2 million reads per sample. Reads were aligned against the phase II reference of the 1000 Genomes Project including decoy sequences d5 (ftp://ftp.1000genomes.ebi.ac.uk/vol1/ftp/technical/reference/phase2_reference_assembly_sequence/hs37d5.fa.gz) using STAR (v2.5.2b)[58] in a 2-pass mapping mode:

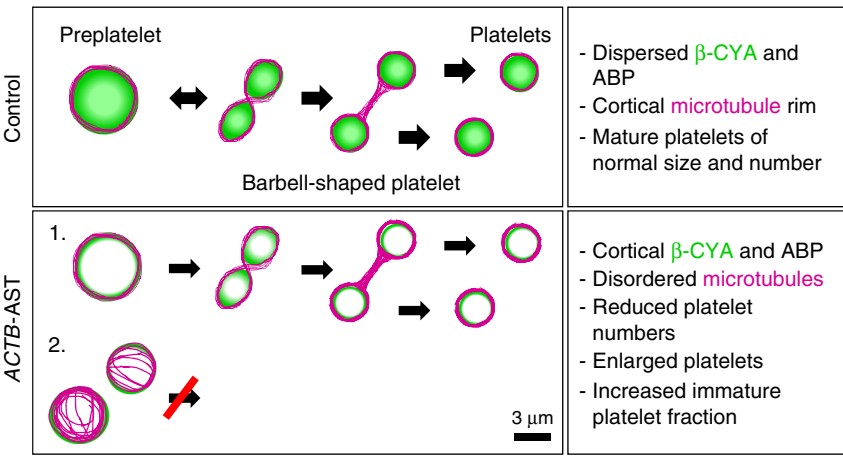

**Fig. 9** Proposed model of the mechanisms underlying thrombocytopenia in *ACTB*-AST. Preplatelets and barbell-shaped platelets shed into the blood stream convert into platelets in a microtubule-dependent manner. A thick band of microtubules is responsible for twisting the center of the preplatelet to form the barbell-shaped platelet, essentially dividing it into two individual terminal platelets. Abscission of terminal platelets subsequently occurs in the circulation. In *ACTB*-AST patients, cytoplasmic actin isoforms and select ABPs (α-actinin 1, NM-2A and to a lesser extent Filamin A) enrich at the preplatelet cortex. The cortical microtubule band observed in mature control cells forms in fewer *ACTB*-AST platelets. Instead, microtubules are highly disordered. We propose that the final platelet processing steps are therefore restricted in preplatelets with a disordered microtubule cytoskeleton, leading to reduced numbers of enlarged and immature platelets in patient circulation

first, an index was created using the genome sequence and gene annotation (Gencode GRCh37.p13 comprehensive gene annotation), against which all reads were aligned. Second, all detected splice junctions of all samples were merged and used as a guide for the second mapping step. Read counts of all annotated genes were extracted from the alignments using the featureCounts function of the Rsubread package (v1.20.6)[59] and genes with 0 counts for all samples were discarded. DESeq2 (v1.10.1)[60] was used to find differentially expressed genes using standard parameters. Triplicates from P4 and P5 were compared against control triplicates separately and as a combined group (P4 and P5) against the control (Supplementary Data 4–6). The differentially expressed genes were filtered using the adjusted *p*-value ≤ 0.05. Significantly upregulated and downregulated genes were chosen based on the Benjamini-Hochberg adjusted *p*-value ≤ 0.01. Principal components analysis was done using the R stats package.

A subset of 213 genes encoding for actin isoforms (6 genes) and known ABPs (207 genes, based on the ref. [3]) was compiled and analyzed further (Supplementary Data 7). Of the 207 ABPs genes assessed, 44 were found to be significantly deregulated in all three analyses (P4 vs. control, P5 vs. control and *ACTB*-AST patients combined vs. control). We revised this list by eliminating the 7 genes with the lowest raw and normalized counts (base mean < 150; Supplementary Data 8). This list includes genes that are not expected to be expressed in this tissue. All shortlisted deregulated genes were then cross-checked on UniProt (http://www.uniprot.org/) and PubMed (https://www.ncbi.nlm.nih.gov/pubmed/) platforms for known disease-causing mutations associated with thrombocytopenia, where platelets are enlarged, as this is a distinguishing clinical feature of *ACTB*-AST patients.

Whole-transcriptome sequencing experiments were designed and performed to screen actin isoform transcription (including that of the mutated *ACTB* allele) and to select a subset of ABP's for further analysis at the protein level. Further conclusions about the impact of these *ACTB* mutations are unable to be drawn from the transcriptome analysis due to the lack of additional healthy control biological replications.

**Cell lysis and western blot analysis.** Cells grown to 70–90% confluency were harvested. Pelleted cells were resuspended, washed in cold 1× dPBS and lysed in ice cold lysis buffer (20 mM Tris-HCl pH 8.0, 100 mM NaCl, 1 mM EDTA, 0.1% NP-40, complete mini protease inhibitor tablet and phosSTOP tablet (both from Roche Applied Science, Penzberg, Germany)) at a final concentration of $1 \times 10^4$ cells per µl of lysis buffer. Lysates were incubated for 30 min on ice, sonicated and vortexed. Total protein concentration was normalized with the Bradford Assay (Bio-Rad, Hercules, CA, US) and readjusted using a Coomassie Protein Assay. Approximately 10–15 µg total protein was separated on polyacrylamide gels (8, 10, or 15%, depending on protein MW) by electrophoresis (35 mA, 45 min) and subsequently transferred onto 0.1 µm pore nitrocellulose membrane (14 V, 1.5 h). Transfer efficiency and protein loading were estimated by Ponceau S staining. Membranes were blocked for 1 h in 5% (w/v) skim milk in TBS-Tween20 (TBS-T). Based on the Ponceau S staining and pre-stained marker, membranes were cut into 2–4 segments and probed with antibodies for proteins running in the respective size range (Supplementary Fig. 6 and Supplementary Fig. 8). Primary antibodies

were incubated overnight in blocking solution at 4 °C (see Antibodies section below and Supplementary Table 2 for dilutions used). Following two TBS-T washes, membranes were incubated in secondary antibody in blocking solution for 1 h and washed thrice in TBS-T prior to developing. Signals were developed with the SuperSignal™ West Femto Maximum Sensitivity Substrate (Thermo Scientific) and images were obtained with the ChemiDoc MP Imaging system using ImageLab software (Bio-Rad). Results were obtained from six independent lysates of varying cell passages. For quantification, each protein of interest was probed at least twice in three different lysates. β-CYA, γ-CYA, pan-actin and β-tubulin were assessed at least once in all lysates. Analysis of western blot signals was performed with Image J software (NIH, Bethesda, ML, US). Values were first adjusted to the total protein content, as determined by Ponceau S staining, and patient samples were normalized to their respective control. Original western blots corresponding to Fig. 4b and Fig. 5b are displayed in Supplementary Fig. 13 and Supplementary Fig. 14, respectively.

**Immunofluorescence microscopy of primary fibroblasts.** For immunofluorescence experiments, cells were seeded at a density of $2 \times 10^4$ on 10 mm glass coverslips in 24-well plates and cultured for 48 h prior to collection. Cells were washed twice with pre-warmed DMEM/F-12 medium and fixed for 30 min with 1% PFA diluted in DMEM/F-12. Following two washes with 1X dPBS, cells were permeabilized with ice cold methanol for 5 min. Samples were washed twice with 1X dPBS (Gibco®; Thermo Scientific) and blocked for 1 h at 23 °C in 2% BSA/dPBS blocking solution prior to antibody labeling. Primary and secondary antibodies were diluted in blocking buffer, as shown in Supplementary Table 2, and were applied for 1 h and 30 min, respectively, with two intermittent blocking buffer washes. Samples were washed twice with 1× dPBS, counterstained with DAPI for 5 min at RT, washed in 1× dPBS, rinsed with ddH$_2$O and mounted in ProLong™ Gold Antifade Mountant (Thermo Scientific). Z-stack images (0.3 µm slices) were obtained with the Leica TCS SP8 Confocal Microscope (Leica Microsystems, Solms, Germany) using a 63× Oil/1.4NA objective. Image analysis was performed using Image J software (NIH, Bethesda, ML, US). Cell and nuclear boundaries were determined manually with ROI selection tools. The mean fluorescence intensity of each channel was calculated from Sum Slice image projections. For overlap coefficient analysis, the 'Just Another Colocalization Plugin' was utilized[61].

**In silico modeling of β-CYA$^{P4/P5}$ and ABP interactions.** Actin models of β-CYA$^{P4/P5}$ were built with Yasara (version 17.8.15, YASARA Biosciences GmbH, Germany) based on PDB structure 5JLH[40]. For β-CYA$^{P4}$, C-terminal residues after I330 were removed. The frameshift peptide was built via homology modeling using PDB structures as templates. The resulting peptide was placed manually into the model and was connected by the Yasara loop building algorithm. After energy minimization, an α-backbone constraint MD-simulation was performed with Desmond 11 MD package, version 2017–4, as distributed by Schrödinger[62]. The β-CYA$^{P5}$ model was generated by deleting the residues spanning between Y337 and S344 and was subsequently closed with a VMD loop building algorithm. After energy minimization an α-backbone constraint MD-simulation was performed. To confirm C-terminus stability, a 50 ns MD-simulation without constraints was

performed with Desmond, using PDB 5JLH as a template. For the MD-simulations the OPLS3 force field was used in a minimized 10 Å orthorhombic water box with 0.15 M sodium chloride. The simulations were performed with a TIP3P water model including a 0.03 ns quick relaxation. The simulation parameters were 300 K and 1 ATM pressure. To confirm the orientation of α-actinin binding on F-actin, the calponin-homology domain of PDB structure 3LUE[63] was used in combination with the F-actin dimer from PDB structure 5JLH[40]. Frodock 2.0 software was used for docking[64]. To confirm stable binding, a 25 ns MD-simulation without constraints was performed with Desmond. MD-simulation results were analyzed with VMD version 1.9.3[65].

**MK differentiation from peripheral blood PBMCs.** For MK and platelet experiments, 15–25 ml of whole blood was collected by venipuncture from two healthy controls, P3, P4 and P5 in the presence of 1.6 mg/ml EDTA. As access to patient blood was limited, these experiments were performed once ($n = 1$), with all samples processed in parallel. MKs were differentiated from whole blood samples using a protocol adapted from Balduini et al.[66] Briefly, PBMCs were isolated from using Lymphosep (C C Pro, Oberdorla, Germany). Up to $2 \times 10^6$ PBMCs were seeded per well in 12-well plates (TPP Techno Plastic Products AG, Trasadingen, Switzerland) with StemSpan ACF medium (StemCell, Vancouver, Canada) supplemented with 10 ng/ml TPO, FLT3-L, IL-6, and IL-11 (all cytokines from Peprotech, Hamburg, Germany). On day 4, APEL2 medium (StemCell) supplemented with 5% Protein-free Hybridoma Medium II (PFHMII, Gibco by Life Technologies, Darmstadt, Germany) and the same cytokine composition was used. On days 1 or 2, 7, 10, and 14, cell morphology was assessed by microscopy and phenotype was analyzed by flow cytometry. For flow cytometry, cells were blocked with FcR-blocking reagent (Miltenyi Biotec, Bergisch Gladbach, Germany) and stained with CD42a-PE (BD Biosciences, Heidelberg, Germany), CD41-APC/Cy7, and CD61-APC (both Biolegend, San Diego, USA). Analysis was performed using a FACS Canto II and FACSDiva software v8.0.1 (BD Biosciences).

**Immunofluorescence microscopy of differentiated MKs.** At day 14, cells were collected from 12-well plates and seeded onto 15 mm coverslips (1.5 thickness) that had been washed with 1 M HCl, 100% Ethanol and coated with poly-l-lysine (Sigma Aldrich, Munich, Germany). Samples were incubated for 24 h at 37 °C and fixed directly in 4% PFA in dPBS. For actin isoform staining, three coverslips per condition were prepared as described in the Immunofluorescence microscopy of primary fibroblasts section above. For β-tubulin labeled samples, cells were permeabilized with 0.1% Triton X-100 for 5 min, blocked in 2% BSA/dPBS blocking solution, incubated with the mouse monoclonal IgG$_1$ primary β-tubulin antibody, washed and probed with the respective secondary goat-anti-mouse IgG$_1$ AlexaFluor488 secondary antibody (see Supplementary Table 2). Samples were washed with dPBS, blocked in 5% normal mouse serum for 30 min and post-labeled with an Alexa Fluor® 647-conjugated anti-CD61 antibody. Samples were imaged with the Leica TCS SP8 confocal. For quantitative assessment of β-tubulin/CD61 labeled MKs, blinded image analysis of 11 images from 3 coverslips per condition was performed.

**Immunofluorescence microscopy of peripheral blood platelets.** Platelet-rich plasma was collected from whole blood via centrifugation at $150 \times g$ for 15 min. Platelets were enriched by centrifugation at $1000 \times g$ for 10 min, washed twice with 1X dPBS and fixed in suspension with 4% PFA for 20 min. Samples were resuspended in 1× dPBS and seeded onto poly-L-lysine coated coverslips for 90 min at 37 °C. Platelets were stained as described in the Immunofluorescence microscopy of primary fibroblasts section above. Samples were imaged with the Leica TCS SP8 confocal using the 63×/1.4NA objective and a zoom factor of 2.5. For platelets size, particle analysis was performed on binary images after intensity thresholding. Overlapping events were excluded from the analysis.

**Transmission electron microscopy (TEM) of whole blood.** Ultrastructural analysis was performed after fixation of whole blood sediments in 2.5% glutaraldehyde for 48 h at 4 °C, post fixation in 1% osmium tetroxide, and sample embedding in araldite. Semi-thin sections were used to identify platelets (thrombocytes) and respective ultrathin sections were stained with uranyl acetate and lead citrate. An EM 902 electron microscope (Zeiss, Oberkochem, Germany) was used to analyze the specimens at 80 kV.

**Antibodies.** Primary monoclonal antibodies used to detect actin isoforms by immunofluorescence and western blot, including the mouse IgG$_1$ anti-β-CYA (clone 4C2), mouse IgG$_{2b}$ anti-γ-CYA (clone 2A3), mouse IgG$_{2a}$ anti-α-SMA (clone 1A4) and mouse IgG$_1$ anti-γ-SMA (clone 20D2, were provided by Prof. Christine Chaponnier. Commercially available unconjugated primary antibodies used include rabbit polyclonal pan-actin (Cell Signaling Technology, Beverly, MA, US; #4968S), mouse anti-β-tubulin (Santa Cruz Biotechnology, Santa Cruz, CA, US, #sc-58880), rabbit anti-NM-2A (Covance, Berkeley, CA, US; #PRB-440P), mouse anti-Filamin 1 (Santa Cruz Biotechnology; #sc-17749), rabbit anti-Filamin A (Abcam Inc, Cambridge, UK; #Ab76289), rabbit anti-α-actinin 1 (Thermo Scientific; #PA5-17308), mouse anti-α-actinin 1 (Merck Millipore, Darmstadt, Germany; #MAB1682), rabbit anti-TPM4 (Merck Millipore; #AB5449) and rabbit anti-

DIAPH1 (Abcam Inc.; #Ab96784). Conjugated primary antibodies used for immunofluorescence or flow cytometry analysis include mouse anti-CD61-AlexaFluor 647 (Biolegend, San Diego, CA, US; #336408), mouse anti-CD61-APC (Biolegend; #336412), mouse anti-CD42a-PE (BD Biosciences, San Jose, CA, US; #558819) and mouse anti-CD41-APC/CY7 (Biolegend; #303716). HRP-conjugated secondary antibodies for western blot analysis were purchased from Thermo Scientific and all fluorescent-conjugated secondary antibodies for immunofluorescence were obtained from Jackson Immunoresearch (West Grove, PA, US). See Supplementary Table 2 for a complete list of antibody clone numbers, product numbers, applications and dilutions.

**Statistical analysis.** Unless stated otherwise, GraphPad Prism 7.0 (GraphPad Software, San Diego, CA, US) was utilized for the graphical representation and statistical analysis of cell-based data. For statistical analysis, the Gaussian distribution of data was first assessed with the D'Agostino & Pearson normality test. In Fig. 4j, where only two conditions were compared, a two-tailed Mann-Whitney test was performed, as data are not normally distributed. For all other experiments (where three or more conditions are compared), normally distributed data sets were analyzed with an ordinary one-way ANOVA followed by Holm-Sidak's multiple comparisons test. Data that did not follow a Gaussian distribution or where the sample size was too small to test the distribution were treated with the Kruskal–Wallis test followed by Dunn's multiple comparisons test. In all cases, $*p < 0.05$, $**p < 0.01$, $***p < 0.001$, and $****p < 0.0001$. Normalized intensity data are shown as mean ± standard deviation (s.d.). All other data are represented as mean ± s.d., mean ± standard error of the mean (s.e.m.) or median with interquartile ranges (IQR) as indicated in the respective figure legends.

## Data availability

DNA- and RNA-Seq data is deposited in the Sequence Read Archive (SRA) under the project ID: PRJNA485028 (SRA ID: SRA755413). Accession numbers SRR7657940, SRR7657941, SRR7657938 represent RNA-Seq triplicates for patient 5, SRR7657942, SRR7657943, SRR7657937 RNA-Seq triplicates of patient 4 and SRR7657939, SRR7657944, SRR7657945 are RNA-Seq triplicates of the healthy control individual. DNA-Seq reads are available under accession numbers SRR7802037, SRR7802031, SRR7802036, SRR7802030, SRR7802034 for patients 1, 2, 3, 4, 5 respectively. DNA-Seq reads from the studied healthy family members in Families A-D are deposited under accession numbers SRR7802032, SRR7802029, SRR7802035, and SRR7802033. DNA-Seq data from patient 6 and other relevant data are available from the authors upon request.

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

## Acknowledgements

We wish to thank the patients and their families, as well as the physicians and genetic counselors who referred them. We thank Christof Litschko, Institute for Biophysical Chemistry, Hannover Medical School for providing excel files for the evaluation of cell migration experiments and discussing the representation of cell migration data. Further, we acknowledge Dr. Rudolph Bauerfeind and the Laser Microscopy Core Facility at Hannover Medical School, and Dr. Andreas Pich and the Proteomics Core Facility at Hannover Medical School. We acknowledge Alexander Krüger for assistance with RNA-Seq. Flow cytometry was performed with the support of the Flow Cytometry Core Facility, a core facility of BIOTEC/CRTD at TU Dresden. N.E. is a participant in the Berlin Institute of Health Charité Clinician Scientist Program, funded by the Charité—Universitätsmedizin Berlin and the Berlin Institute of Health. This work was financially supported by DFG grants MA 1081/22-1 and MA 1081/23-1 (D.J.M); Volkswagen Foundation grant VWZN3012 (D.J.M. and M.H.T); DFG grant DI 2170/3-1 (N.DD).

## Authors contributions

S.L.L. conceived, performed, analyzed and interpreted all cell experiments. N.DD. and N. E. defined *ACTB*-AST as a novel clinical entity. P.Y.A.R., M.H.T. and D.J.M. conceived,

performed and analyzed structural studies. D.E. and C.F. performed in vitro mega-karyocyte differentiation and contributed to the interpretation of these experiments. T.R. performed blind analysis of megakaryocyte data. W.S. performed TEM of patient blood samples. N.E., R.K., M.C.F., M.J.L., M.J.F., J.A.L., K.B., T.M.N., E.S., and A.R. recruited patients, collected clinical data, and analyzed patient sequencing results. M.C.F., R.H. and R.K. analyzed blood and bone marrow smears and defined hematological phenotype of *ACTB*-AST. L.G. performed flow cytometry, I.N., K.S., K.G., and B.K. performed and analyzed whole-transcriptome sequencing, C.C. provided actin monoclonal antibodies and critical discussion for data interpretation. N.DD. coordinated the study and with D.J. M. co-sought funding. S.L.L., N.DD. and D.J.M. wrote the manuscript. All other authors critically analyzed and edited the manuscript.

## Additional information

**Competing interests:** The authors declare no competing interests.

