## [Peer Review File · Nature Communications]

Reviewer #1 (Remarks to the Author):

The manuscript by Latham et al. reports on 6 patients with heterozygous variants in ACTB gene encoding beta-cytoplasmic actin. These patients exhibit low platelet count with platelet anisotropy, associated with microcephaly. By growing patient's dermal fibroblasts, the authors provide evidence that beta-CYA variants impair a number of actin-linked cell processes such as cell morphology, spreading and migration. They show a modification of actin isoforms expression, notably a compensating expression of alpha-SMA in patients which is not expressed in control cells. Other cytoskeleton proteins are overexpressed notably myosin-2A. Both CYA and myosin distribute abnormally in platelets from these patients, suggesting that it may be responsible for a decrease in platelet production. The authors also propose a model of how mutations may affect myosin and alpha-actinin interaction. This manuscript is interesting as it reports for a novel actinopathy leading to general abnormalities in cell cytoskeleton. However some points need to be clarified.

In the text, page 2, the authors write "Heterozygous germ-line mutation in both genes have been associated with BWCF,". And after "The consequences of ACTG1 germ-line mutations are less pleiotropic". This is not clear, the same ACTG1-germ-line" mutation cannot lead to pleiotropic and less pleiotropic consequences.

Page 4, second paragraph: "gama-CYA levels are reduced" and "compensatory increase in gama-CYA". This is not clear. The authors should provide immunofluorescence images showing both CYAs in whole cells in the same figure 3a rather than in supplemental data, so as to be able to directly compare with the control cells shown in Fig.3a (same scale and image acquisition parameters).

Fig.3c shows that total actin is unchanged using pan-actin antibody and WB experiments on fibroblasts. This is in sharp contrast to Suppl. Fig.5b where again by WB the authors show a total absence of alpha-SMA in control fibroblasts while it is strongly expressed in patients' fibroblasts (P3 and P5). Explain these discrepancies.

The authors proposes a scenario for the thrombocytopenia based on the fact that cytoskeleton anomalies often lead to defective platelet formation and production. The authors should provide transmission electron microscopy images of washed platelets from at least one patient and control to evidence whether the cytoskeletal defects impact the ultrastructure platelets, a known feature of many patients' platelets having defective cytoskeleton.

One weakness of the study is the lack of evidence directly relating actin mutations to the patients' phenotype. Hence, providing evidence that peripheral CD34+-derived MK from patients have indeed defective proplatelet formation would strongly strengthen the conclusions of the study in terms of direct pathophysiological consequence of the mutations.

Reviewer #2 (Remarks to the Author):

This manuscript by Latham et al describes the identification of 4 new mutations in the cytoplasmic beta actin gene and the further characterisation of 2 of these. They report a phenotype in these patients very similar to that of BWCF, but with additional macrothrombocytopenia.

If correct, then to my knowledge, this is the first report of actin mutations causing thrombocytopenia. Whilst this is important, the claim that the mutation is responsible for the thrombocytopenia via changes in ABPs and this “has resolved key components of a cytoskeletal filament population that both generates and transmits the forces required during the final stages of thrombopoiesis” is overstated.

Generally the experiments appear well designed and carried out appropriately. There are however, several areas that need more details or explanation.

1. How were patients selected? There is no mention of the criteria for selection for WES.
2. The data for platelet number and size is confusing (Supplementary table 1). What does gpt/l stand for? Why is the reference range for each patient reported as being different? What are the mean platelet volumes for P1 – 5 and the BWCF platelets?
3. There is no detail or discussion of how many variants/mutations were identified in the WES. On what criteria were the others rejected as being not-relevant? Can the authors be sure that other variants/mutations are not playing a role/causative effect?
4. The patient fibroblasts used in Figure 2a clearly show an altered morphology phenotype at low density with P5 cells lacking clear polarisation. However, at high density the general phenotype is very similar (lower panels) with all cells becoming polarised and cell proliferation appears higher in the P3 and P5 cells. The authors do not discuss this. What might the relevance of this be?
5. In Figure 2b and 2c – at what density were cells measured? One might expect different results dependent on cell density as suggested by their fig 2a and might give an insight as to the function of the mutation.

6. In Figure 2b and c, what numbers of cells were counted? Presumably, as this one patient sample, there is only 1 replicate? This question of replicates also comes up in Figure 3C (and also 4b). In the western blots the authors report $n = 9$, but there is only 1 patient for each mutation. How can this be $n = 9$? A similar issue is present for the fibroblast staining experiments where conclusions are drawn on a single replicate.

7. Figure 2D is not referred to in the text.

8. Although the images in figure 3a appear to show different patterns of staining for the two actin isoforms, how specific are the beta and gamma actin antibodies used. Supplementary table 8 gives no supplier information or reference for these antibodies. Given that the two proteins only differ by 4 amino acids, how confident are the authors of the specificity? They should either show the data supporting this or provide references to the antibodies.

9. The analysis of actin and ABP distributions between the different cell lines is an impressive undertaking, but is quite subjective, with only a few cells presented for each phenotype. The authors should look to quantify the differences somehow, or classify the phenotypes over a large number of cells to show how representative the images are and how robust are the differences reported. This would help increase confidence in a subjective measure on a single replicate.

10. The modelling data is impressive and the possible effects of the mutations on the actin monomer structure and putative interactions is well described. The analysis of gene expression is also interesting showing up regulation of ABPs. However, there is a well characterised role of actin as a regulator of gene expression via the MAL-SRF and it is likely that these actin mutations may be simply affecting expression of ABPs via this method. The authors should consider this.

11. A major limitation of the work is the assumption that the increase in amount of ABPs is responsible for the thrombocytopenia. It is a large jump from a fibroblast to a megakaryocyte and it is clear that the role of actin and ABPs in the process of platelet production is complex and not fully understood. To be able to make this claim, the authors would need to characterise these mutations in megakaryocytes.

Reviewer #3 (Remarks to the Author):

Mutations in the beta-actin encoding gene ACTB have hitherto been causally associated with a range of autosomal dominant disorders featuring developmental/neurological abnormalities, of which the most severe is Baraitser-Winter Syndrome (Proaccio et al, 2006; Riviere et al, 2012). In addition, an isolated report from 1999 describes a case carrying a variant encoding a Glutamic acid to Lysine substitution at amino acid (AA) 364 of 375 in ACTB having thrombocytopenia, intellectual disability and an immune phenotype (Nunoi et al, 1999). These and other reports indicate that rare variants in ACTB are associated with a relatively wide spectrum of phenotypes.

In keeping with the paper from 1999, the authors of the present manuscript write that other rare coding variants in the 3' end of ACTB are responsible for a neurological phenotype and thrombocytopenia. Although in Nunoi et al only moderate intellectual disability was reported, here the authors report microcephaly in 5/6 cases. The authors report four new rare variants: one missense variant at AA 313, two frameshift variants at AAs 331 and 368 and an in-frame deletion at AA 338.

Thus, it appears that variants causal of Baraitser-Winter syndrome are located in exons 2-4 while, based on the present work and that of Nunoi et al, variants located in exons 5-6 can cause thrombocytopenia and microcephaly.

Assuming that these 3' variants are genuinely associated with syndromic thrombocytopenia, the key question that needs to be addressed is why this is the case, particularly given that variants causal of Baraitser-Winter apparently do not cause thrombocytopenia. The authors have performed a series of cell-based and transcriptomic assays on fibroblasts obtained from two patients and a control. Although the experimental data are informative of the defect in fibroblasts, the mechanism underlying the defect in platelet formation (or clearance) remains elusive, not least because there are no data on cells relevant to the thrombocytopenia (e.g., megakaryocytes). Furthermore, there are no data on Baraitser-Winter fibroblasts to help explain the relationship between the location of the ACTB variants and the phenotype.

Platelets are readily accessible cells and their progenitors (megakaryocytes) can be generated using blood-derived stem cells (or alternatively genetically modified CD34 or iPS stem cells). In the current manuscript, the thrombocytopenia remains unexplained and it is not even known whether the platelets are functional. It is very hard to put forward a new disease entity comprising thrombocytopenia without any detailed studies to illustrate or explain this. RNA sequencing of skin fibroblast to try to explain a mechanism relevant for megakaryopoiesis and platelet formation is an important shortcoming. It is known that many proteins that alter this process are very specifically expressed in megakaryocytes and platelets or have a specific function in proplatelet formation that doesn't exist in other cells, as this process is highly unique.

Further to this, I have two specific major comments on the data analysis:

1) Genetic analysis

There is practically no description of the analytical or statistical methodology used to declare these rare variants as being likely associated with the phenotype. The Methods section states that "NextGENe® software (SoftGenetics, LLC, State College, PA), Cartagenia Bench Lab NGS software (Agilent Technologies), or in-house software integrating open-source tools, such as BWA, GATK, FreeBayes toolkit, Jannovar and PhenIX, were used" but there is no explanation as to how any of these tools were actually used and for what purpose and how their use contrasts with the use of other similar software mentioned earlier on. For example, the authors state that the BWA aligner was used just after stating that CLC Biomedical Genomics Workbench was used for alignment.

There is a marked absence of pedigree data, despite the fact that two of the variants were observed in two relatives each. Co-segregation analysis would be an important addition to this work.

The GnomAD database from the Broad Institute lists the following rare variants that are also in the 3' end of ACTB, one of which affects the same residue as present in the case described by Nuno et al:

p.Lys336Arg

p.Ser350Thr

p.Leu349SerfsTer29

p.Glu364Gln

p.Ser365Thr

p.Arg372Cys

Can the authors speculate on why these variants are or are not causal? Is it possible that some ACTB-related thrombocytopenia and microcephaly cases are present in the GnomAD database or do the authors believe these GnomAD variants have different effects from the ones the authors are reporting?

2) RNA-seq analysis

Three separate libraries were prepared from one P3, one P5 and a single control fibroblast culture (i.e. from three separate individuals).

Differential expression analysis was then performed comparing P3 and P5 libraries (jointly and singly) to control libraries. The analysis as described is problematic because there is no biological replication of controls and hence the inter-individual variance cannot be properly estimated. An example might help elucidate the problem: suppose we compared the three technical replicates from P5 to the three technical replicates from P3. Clearly, all genes will be truly differentially expressed between the two (irrespective of power to detect this difference) due to the differences (genetic and environmental) between the two individuals. What is important is whether the population of patients have a different mean expression relative to a population of controls. Yet here, there is no biological replication allowing this question to be addressed. Furthermore, in the analysis using both P3 and P5 data, the variance due to technical replication is not distinguished in the modelling from the variance due to biological replication amongst the two cases, yet these are two distinct sources of variability.

There is also a fundamental problem with the way in which the data for Figure 4A were processed. The authors state that "genes were eliminated from the analysis when < 300 raw counts were observed in any of the samples analyzed." In addition to the fact that normalised read counts, not raw counts, should be used to avoid bias against conditions for which slightly fewer reads were generated, this approach will generate a bias in favour of genes up-regulated in patient samples because there are more patient libraries than control libraries (6 vs 3). It stands to reason that the more libraries there are from a particular condition, the higher the chance that at least one of them dips below the threshold.

In general, the analysis suffers from the authors' use of simple differential expression methodology to attempt to classify genes by pattern of expression (as attempted in the pie chart in Figure 4A). Principled methods for polytomous model selection using RNA-seq exist that could help.

As a final comment, the motivation for the RNA-seq work is not clearly stated: what is the hypothesis that the experiments are designed to assess? Why would an abnormal ACTB protein in the cytoplasm lead to differential transcript levels of actin-binding proteins specifically? Should the authors not make a distinction between thrombocytopenia-related ABPs that are implicated in disease through a gain-of-function mechanism (e.g., DIAPH1) from those acting through other mechanisms (e.g., ACTN1, TPM4)?

Reviewers' comments

Megakaryocyte/Thrombocyte response for all Reviewers

As our patients are geographically dispersed throughout Germany (Berlin, Augsburg and Munich) and the USA, access to fresh blood samples is limited. For this reason, our initial manuscript submission utilized patient-derived primary dermal fibroblasts (from P4 and P5) and peripheral blood smears (from P1 and P5) to study the effects of *ACTB*-TMS variants. Following the request by all reviewers for additional experiments in megakaryocytes and thrombocytes, Family B (P3 and P4) and Family C (P5) patients gave consent for an additional peripheral blood donation. Due to the low frequency of CD34+ stem/progenitor cells (~0.1%) in peripheral blood without G-CSF mobilization and the low sample volume that could be obtained from patients (10-20 ml), we adapted a megakaryocyte differentiation protocol using the complete PBMC population. With these samples we evaluated the effect of two *ACTB*-TMS mutations on various aspects of thrombopoiesis (Figure 8, Supplementary Figures 10 and 11). Additionally, we improved the immunofluorescent assessment of cytoskeletal constituents in thrombocytes (Figures 2 and 7) and assessed platelet ultrastructure by transmission electron microscopy (Supplementary Figure 9).

The additional experiments show that these *ACTB*-TMS variants have no effect on proplatelet length or branching. However, we observe a significant effect on the organization of microtubules within proplatelet swellings and purified thrombocytes. Based on our findings, we propose that inhibition of the final stages of platelet maturation causes enlarged platelets in *ACTB*-TMS patients. Our data are in agreement with work from the Italiano lab (Harvard Medical School) and other groups in the field, that actin and microtubule forces are essential for the maturation of platelets from proplatelet precursors. We would like to thank the reviewers for their recommendation that we perform these experiments.

Reviewer #1 (Remarks to the Author):

The manuscript by Latham *et al.* reports on 6 patients with heterozygous variants in *ACTB* gene encoding beta-cytoplasmic actin. These patients exhibit low platelet count with platelet anisotropy, associated with microcephaly. By growing patient's dermal fibroblasts, the authors provide evidence that beta-CYA variants impair a number of actin-linked cell processes such as cell morphology, spreading and migration. They show a modification of actin isoforms expression, notably a compensating expression of alpha-SMA in patients which is not expressed in control cells. Other cytoskeleton proteins are overexpressed notably myosin-2A. Both CYA and myosin distribute abnormally in platelets from these patients, suggesting that it may be responsible for a decrease in platelet production. The authors also propose a model of how mutations may affect myosin and alpha-actinin interaction. This manuscript is interesting as it reports for a novel actinopathy leading to general abnormalities in cell cytoskeleton. However some points need to be clarified.

In the text, page 2, the authors write "Heterozygous germ-line mutation in both genes have been associated with BWCF,". And after "The consequences of *ACTG1* germ-line mutations are less pleiotropic". This is not clear, the same *ACTG1*-germ-line" mutation cannot lead to pleiotropic and less pleiotropic consequences. *We apologize for the miswording of this section. We had intended to highlight that *ACTG1* is associated with less severe and fewer clinical phenotypes compared to *ACTB*. This is now addressed more carefully in the third paragraph of the introduction.*

Page 4, second paragraph: "gamma-CYA levels are reduced" and "compensatory increase in gamma-CYA". This is not clear. The authors should provide immunofluorescence images showing both CYAs in whole cells in the same figure 3a rather than in supplemental data, so as to be able to directly compare with the

control cells shown in Fig.3a (same scale and image acquisition parameters). Regarding the text, we removed the sentence about γ -CYA levels being reduced in this filament population. For the images requested, we have now consolidated the control, P4 and P5 fibroblast images showing β -CYA and γ -CYA distribution into one figure. However, we have kept this figure in the supplementary material as it is in line with published reports (Dugina *et al.* 2009, *JCS*; Latham *et al.* 2013, *FASEB J*), does not give anything additional to our story and takes up considerable figure space. As in the previous version, all images were collected on the same day, with the same acquisition parameters and are shown at the same scale.

Fig.3c shows that total actin is unchanged using pan-actin antibody and WB experiments on fibroblasts. This is in sharp contrast to Suppl. Fig.5b where again by WB the authors show a total absence of α -SMA in control fibroblasts while it is strongly expressed in patients' fibroblasts (P3 and P5). Explain these discrepancies. Double the amount of protein was required to detect α -SMA in these samples. We apologize that this was not indicated in the original version of the manuscript. As the original representative western blot images for the actin isoforms were also not from the same lysate preparations, we have now replaced the β -CYA and γ -CYA images to be consistent with the α -SMA. Additionally, the α -SMA image has been moved to Figure 4b. We have now indicated in the figure and the figure legend that the protein loading was doubled for α -SMA detection.

The authors proposes a scenario for the thrombocytopenia based on the fact that cytoskeleton anomalies often lead to defective platelet formation and production. The authors should provide transmission electron microscopy images of washed platelets from at least one patient and control to evidence whether the cytoskeletal defects impact the ultrastructure platelets, a known feature of many patients' platelets having defective cytoskeleton. We performed transmission electron microscopy of samples from a healthy control, P3, P4 and P5 (Supplementary Figure 9). Our assessment showed that all regular intracellular content in the form of dense granules, α -granules, mitochondria and lysosomes is present in both healthy control and *ACTB*-TMS patient platelets. We have included representative images from each condition and a brief description of these data in the Supplementary material. Follow up studies are required for further in depth characterization of *ACTB*-TMS platelet ultrastructure.

One weakness of the study is the lack of evidence directly relating actin mutations to the patients' phenotype. Hence, providing evidence that peripheral CD34⁺-derived MK from patients have indeed defective proplatelet formation would strongly strengthen the conclusions of the study in terms of direct pathophysiological consequence of the mutations.

We appreciate this suggestion, which was also made by the other reviewers. Therefore we have addressed it in a joined response at the beginning of the rebuttal letter.

Reviewer #2 (Remarks to the Author):

This manuscript by Latham *et al.* describes the identification of 4 new mutations in the cytoplasmic beta actin gene and the further characterisation of 2 of these. They report a phenotype in these patients very similar to that of BWCF, but with additional macrothrombocytopenia. If correct, then to my knowledge, this is the first report of actin mutations causing thrombocytopenia. Whilst this is important, the claim that the mutation is responsible for the thrombocytopenia via changes in ABPs and this “has resolved key components of a cytoskeletal filament population that both generates and transmits the forces required during the final stages of thrombopoiesis” is overstated. Generally the experiments appear well designed and carried out appropriately. There are however, several areas that need more details or explanation.

1. How were patients selected? There is no mention of the criteria for selection for WES. All patients received genetic testing as they presented with syndromic developmental delay/intellectual disability and could not be diagnosed with a specific Mendelian disorder. The results and medical records of P1-P4 were sent to N.DD for a second opinion after identification of likely pathogenic variants in *ACTB*, and as patients did not fulfil the diagnostic criteria for Baraitser-Winter-Cerebrofrontofacial syndrome (BWCFF). P5 was diagnosed by N.DD and the data of P6 was discussed with N.DD during the David W. Smith Workshop (August 14-19, 2015 at the Harbourtowne Conference Center in St. Michaels Maryland). Therefore, this cohort was ascertained after the identification of *ACTB* variants in patients with clinical features not consistent with BWCFF. The corresponding explanation has been added to the text.

2. The data for platelet number and size is confusing (Supplementary table 1). What does gpt/l stand for? Why is the reference range for each patient reported as being different? What are the mean platelet volumes for P1 – 5 and the BWCFF platelets? Gpt/L stands for giga-particle counts per liter. This explanation has been added to the legend in Supplementary Table 1. As blood counts were performed in different diagnostic laboratories, we have included their respective reference ranges. These ranges also vary with age and gender. Regarding the mean platelet volume for P1-P5, automated measurements were not possible due to marked anisotropy. Hemocytometers always reported an error. In our new experiments, we have assessed the mean diameter of platelets from P3-P5 with particle analysis using Image J software (Figure 2). These results show a significant increase in patient platelet size compared to the healthy control (values given in Supplementary Table 7). Unfortunately, no exact platelet volume measurements are available for the BWCFF cohort. However, the general count and morphology has always been described as normal in available reports.

3. There is no detail or discussion of how many variants/mutations were identified in the WES. On what criteria were the others rejected as being not-relevant? Can the authors be sure that other variants/mutations are not playing a role/causative effect? We have now provided the information on filter criteria and showed the list of all candidate variants. *ACTB* variants were determined to be likely pathogenic by independent diagnostic laboratories using the ACMG criteria for variant interpretation (Richards *et al.* 2015, *Genet Med*). After re-evaluation of whole exome data for this revision, *ACTB* variants remain the only likely disease-causing mutation (see details in the reply to major concerns). The complete list of other candidate variants is provided in Supplementary Table 8.

4. The patient fibroblasts used in Figure 2a clearly show an altered morphology phenotype at low density with P5 cells lacking clear polarisation. However, at high density the general phenotype is very similar (lower panels) with all cells becoming polarised and cell proliferation appears higher in the P3 and P5 cells. The authors do not discuss this. What might the relevance of this be? Proliferation is not significantly affected in patient fibroblasts (please see the growth curve data in Supplementary Figure 3b). Regarding these points, we have now included the following section into paragraph 4 of the new discussion: ‘The fact that proliferation is normal in fibroblasts suggests that it is likely not a major contributor to disease phenotype expression. In contrast, the cells inability to polarize and move persistently appears to be more critical. Furthermore, the observation that fibroblasts show clear deficits at low confluence and not at high confluence shows how the mutations affect mechanotransduction. Polarization, movement and morphology are greatly affected at the single cell level, whilst the wild-type phenotype is rescued with cell-cell contact formation.’

5. In Figure 2b and 2c – at what density were cells measured? One might expect different results dependent on cell density as suggested by their fig 2a and might give an insight as to the function of the mutation. In the revised manuscript, these figures are now panels 3b and 3c. Beside the top panel of Figure 3a, all other experiments were performed with sub-confluent cells. For the attachment surface area, cells needed to be sub-confluent in order for cell boundaries to be defined. We estimate between 50-70% confluency for all the images analyzed. For flow cytometry, cells were estimated to be 70% confluent. We have incorporated this information into the relevant areas of the text.

6. In Figure 2b and c, what numbers of cells were counted? Presumably, as this one patient sample, there is only 1 replicate? This question of replicates also comes up in Figure 3C (and also 4b). In the western blots the authors report $n = 9$, but there is only 1 patient for each mutation. How can this be $n = 9$? A similar issue is present for the fibroblast staining experiments where conclusions are drawn on a single replicate. For all experiments in the manuscript, we have now included the number of events analyzed in either the figure legend or directly in the figure (indicated in the figures in brackets). Each condition (C, P4 or P5) is one biological sample that is specifically named after the patient. In the western blot figures, N represented the number of times experiments were repeated. As we have removed the western blot tables from the figures, it is now indicated in the figure legend that these are technical replicates. We have tried to clarify this throughout the manuscript.

7. Figure 2D is not referred to in the text.
This has been amended in the text.

8. Although the images in figure 3a appear to show different patterns of staining for the two actin isoforms, how specific are the beta and gamma actin antibodies used. Supplementary table 8 gives no supplier information or reference for these antibodies. Given that the two proteins only differ by 4 amino acids, how confident are the authors of the specificity? They should either show the data supporting this or provide references to the antibodies. These antibodies were provided by a co-author, Professor Christine Chaponnier, and as such we gave the clone numbers but not the supplier information. We have now indicated in Supplementary Table 10 a number of commercial suppliers. We have validated these antibodies in-house with purified β -CYA and γ -CYA recombinant protein and an extensive body of literature has demonstrated their ability to detect discrete filament populations in cells (Dugina *et al.* 2009, *JCS*; Latham *et al.* 2013, *FASEB J*; Marzook *et al.* 2017, Cytoskeleton, etc.). The Dugina *et al.* 2009, *JCS* citation first describing these antibodies has been given in the Supplementary Table and in the Results section of the text.

9. The analysis of actin and ABP distributions between the different cell lines is an impressive undertaking, but is quite subjective, with only a few cells presented for each phenotype. The authors should look to quantify the differences somehow, or classify the phenotypes over a large number of cells to show how representative the images are and how robust are the differences reported. This would help increase confidence in a subjective measure on a single replicate. We have included additional analyses in the actin and ABP related figures (Figure 4d, Figure 5e, Supplementary Figure 8a and Supplementary Figure 11a). The β -CYA: γ -CYA ratio data were included as they demonstrate consistency in actin isoform relationships between the different cell types in P3, P4 and P5. For ABP, we performed intensity analysis of sub-nuclear regions, supporting our observations that α -actinin 1, NM-2A and filamin A are all increasingly incorporated into sub-nuclear filament populations in patient fibroblasts.

10. The modelling data is impressive and the possible effects of the mutations on the actin monomer structure and putative interactions is well described. The analysis of gene expression is also interesting showing up regulation of ABPs. However, there is a well characterised role of actin as a regulator of gene expression via the MAL-SRF and it is likely that these actin mutations may be simply affecting expression of ABPs via this method. The authors should consider this. We agree that the MAL-SRF circuit is one of the possible mechanisms of actin-genome communication in *ACTB*-TMS cells. In our RNA-Seq data, we indeed saw a slight but statistically significant increase of SRF transcription (\log_2FC 0.8), along with two of the aberrantly expressed ABPs, *MYH9* and *FLNA*, which are SRF-MRTF regulated genes. However, taking into account possible bias of the selected RNA-Seq approach (see Reviewer 3 comments) RNA-Seq results were significantly reduced in the revised version of the manuscript. Elucidation of actin-genome cross-talk mechanisms will be a topic of subsequent studies.

11. A major limitation of the work is the assumption that the increase in amount of ABPs is responsible for the thrombocytopenia. It is a large jump from a fibroblast to a megakaryocyte and it is clear that the role of actin and ABPs in the process of platelet production is complex and not fully understood. To be able to make this claim, the authors would need to characterize these mutations in megakaryocytes. Firstly, we would like to clarify; we do not think that it is the increase in ABP amount that leads to thrombocytopenia. We hypothesized that perturbed actin-ABP interactions result in compensatory ABP changes at the transcript and protein levels. RNA-Seq was the screening tool used to produce the ABP shortlist and all genes were cross-checked for thrombocytopenia associations. This was done regardless of whether genes were upregulated or downregulated. It was coincidental that the five candidates linked to thrombocytopenia with enlarged platelets were all upregulated. To refine this shortlist, we checked whether the transcript levels matched the protein analysis. In the case of *Diaph1*, data were inconsistent, and as such it was not assessed further. We have included the Tpm4.2 data in the main figures now to make it clearer that we do not believe that increased ABP levels are responsible for thrombocytopenia. Whilst Tpm4.2 is upregulated at the protein level, it does not localize to our filament population of interest. Our structural analysis shows that affected proteins interact with β -CYA at the mutation affected sites. We have performed characterization of the patient-derived megakaryocytes and thrombocytes. Please see our joint response at the beginning of the rebuttal letter.

Reviewer #3 (Remarks to the Author):

Mutations in the beta-actin encoding gene *ACTB* have hitherto been causally associated with a range of autosomal dominant disorders featuring developmental/neurological abnormalities, of which the most severe is Baraitser-Winter Syndrome (Proaccio et al, 2006; Riviere et al, 2012). In addition, an isolated report from 1999 describes a case carrying a variant encoding a Glutamic acid to Lysine substitution at amino acid (AA) 364 of 375 in *ACTB* having thrombocytopenia, intellectual disability and an immune phenotype (Nunoi et al, 1999). These and other reports indicate that rare variants in *ACTB* are associated with a relatively wide spectrum of phenotypes.

In keeping with the paper from 1999, the authors of the present manuscript write that other rare coding variants in the 3' end of *ACTB* are responsible for a neurological phenotype and thrombocytopenia. Although in Nunoi et al only moderate intellectual disability was reported, here the authors report microcephaly in 5/6 cases. The authors report four new rare variants: one missense variant at AA 313, two frameshift variants at AAs 331 and 368 and an in-frame deletion at AA 338. Thus, it appears that variants causal of Baraitser-Winter syndrome are located in exons 2-4 while, based on the present work and that of Nunoi et al, variants located in exons 5-6 can cause thrombocytopenia and microcephaly.

Assuming that these 3' variants are genuinely associated with syndromic thrombocytopenia, the key question that needs to be addressed is why this is the case, particularly given that variants causal of Baraitser-Winter apparently do not cause thrombocytopenia. The authors have performed a series of cell-based and transcriptomic assays on fibroblasts obtained from two patients and a control. Although the experimental data are informative of the defect in fibroblasts, the mechanism underlying the defect in platelet formation (or clearance) remains elusive, not least because there are no data on cells relevant to the thrombocytopenia (e.g., megakaryocytes). Furthermore, there are no data on Baraitser-Winter fibroblasts to help explain the relationship between the location of the *ACTB* variants and the phenotype. Platelets are readily accessible cells and their progenitors (megakaryocytes) can be generated using blood-derived stem cells (or alternatively genetically modified CD34 or iPS stem cells). In the current manuscript, the thrombocytopenia remains unexplained and it is not even known whether the platelets are functional. It is very hard to put forward a new disease entity comprising thrombocytopenia without any detailed studies to illustrate or explain this. RNA sequencing of skin fibroblast to try to explain a mechanism relevant for megakaryopoiesis and platelet formation is an important shortcoming. It is known that many proteins that alter this process are very specifically expressed in megakaryocytes and platelets or have a specific function in proplatelet formation that doesn't exist in other cells, as this process is highly unique.

We greatly appreciate the suggestion to study reprogrammed megakaryocytes. As similar content was raised by other reviewers we have addressed this point in the joined response at the beginning of the rebuttal letter. In addition, the reviewer referred to experiments with Baraitser-Winter fibroblasts. We think it would be very interesting to understand why Baraitser-Winter variants do not cause thrombocytopenia. However, this would require analysis of a substantial number of different mutations since these variants affect different domains and binding sites. Such an extensive study on many Baraitser-Winter fibroblast cultures was beyond the scope of this study but we hope we can address this in future projects.

Further to this, I have two specific major comments on the data analysis:

1) Genetic analysis

There is practically no description of the analytical or statistical methodology used to declare these rare variants as being likely associated with the phenotype. The Methods section states that "NextGENE® software (SoftGenetics, LLC, State College, PA), Cartagenia Bench Lab NGS software (Agilent Technologies), or in-house software integrating open-source tools, such as BWA, GATK, FreeBayes toolkit, Jannovar and PhenIX, were used" but there is no explanation as to how any of these tools were actually used and for what purpose and how their use contrasts with the use of other similar software mentioned earlier on. For example, the authors state that the BWA aligner was used just after stating that CLC Biomedical Genomics Workbench was used for alignment. Originally, all patients were studied in different diagnostic laboratories in Germany and the USA, with each laboratory using their own strategies for sequencing and data analysis. In this revised version, we reanalyzed patients 1-5 with whole exome sequencing using a uniform capture (IDT xGen Exome research panel), sequencing platform and analysis pipeline. To avoid further confusion, we decided to omit the original tests from the revised manuscript and only provide the whole exome sequencing description. As such, the analytical methodology is more clearly detailed in this revised manuscript (Page 17 of Materials and Methods). This new analysis did not reveal any additional potentially causative variants, further supporting the pathogenicity of *ACTB* mutations. All variants identified are now listed in Supplementary Table 8.

There is a marked absence of pedigree data, despite the fact that two of the variants were observed in two

relatives each. Co-segregation analysis would be an important addition to this work. The family information has been added to the text and pedigree diagrams have been included in Figure 1. Extended family history of P1 and P2, as well as P3 and P4 was unremarkable. Unfortunately, no other family members, e.g. grandparents of P1 and P3, were available for co-segregation analysis.

The GnomAD database from the Broad Institute lists the following rare variants that are also in the 3' end of ACTB, one of which affects the same residue as present in the case described by Nunoi et al:

- p.Lys336Arg
- p.Ser350Thr
- p.Leu349SerfsTer29
- p.Glu364Gln
- p.Ser365Thr
- p.Arg372Cys

Can the authors speculate on why these variants are or are not causal? Is it possible that some ACTB-related thrombocytopenia and microcephaly cases are present in the GnomAD database or do the authors believe these GnomAD variants have different effects from the ones the authors are reporting? Each of the listed variants was observed once among > 246 000 alleles with the frequency below 0.000004, with the exception of p.Lys336Arg, which was seen only in genome data (once in 30920 alleles, frequency 3.234e-5). Considering the mildness of the developmental disability and subclinical course of thrombocytopenia, we hypothesize that these rare variants might have been considered not significant, allowing the carriers to be included in the large sequencing cohorts that are incorporated into the GnomAD database. It also cannot be excluded that these variants are not fully penetrant.

The variant p.Glu364Gln affects the same residue as the mutation described by Nunoi *et al.* However, the substitution of Glu with Lys is likely to have more severe consequences on protein structure and function. We thank Reviewer 3 for bringing this concern to our attention and we have included a paragraph about the possible consequences of GnomAD listed ACTB variants to the Supplementary Notes.

2) RNA-seq analysis

Three separate libraries were prepared from one P3, one P5 and a single control fibroblast culture (i.e. from three separate individuals). Differential expression analysis was then performed comparing P3 and P5 libraries (jointly and singly) to control libraries. The analysis as described is problematic because there is no biological replication of controls and hence the inter-individual variance cannot be properly estimated. An example might help elucidate the problem: suppose we compared the three technical replicates from P5 to the three technical replicates from P3. Clearly, all genes will be truly differentially expressed between the two (irrespective of power to detect this difference) due to the differences (genetic and environmental) between the two individuals. What is important is whether the population of patients have a different mean expression relative to a population of controls. Yet here, there is no biological replication allowing this question to be addressed. Furthermore, in the analysis using both P3 and P5 data, the variance due to technical replication is not distinguished in the modelling from the variance due to biological replication amongst the two cases, yet these are two distinct sources of variability.

There is also a fundamental problem with the way in which the data for Figure 4A were processed. The authors state that "genes were eliminated from the analysis when < 300 raw counts were observed in any of the samples analyzed." In addition to the fact that normalised read counts, not raw counts, should be used

to avoid bias against conditions for which slightly fewer reads were generated, this approach will generate a bias in favour of genes up-regulated in patient samples because there are more patient libraries than control libraries (6 vs 3). It stands to reason that the more libraries there are from a particular condition, the higher the chance that at least one of them dips below the threshold.

In general, the analysis suffers from the authors' use of simple differential expression methodology to attempt to classify genes by pattern of expression (as attempted in the pie chart in Figure 4A). Principled methods for polytomous model selection using RNA-seq exist that could help. As a final comment, the motivation for the RNA-seq work is not clearly stated: what is the hypothesis that the experiments are designed to assess? Why would an abnormal ACTB protein in the cytoplasm lead to differential transcript levels of actin-binding proteins specifically? Should the authors not make a distinction between thrombocytopenia-related ABPs that are implicated in disease through a gain-of-function mechanism (e.g., DIAPH1) from those acting through other mechanisms (e.g., ACTN1, TPM4)? We appreciate Reviewer 3's comments regarding our RNA-Seq analysis and agree that multiple biological replicates would significantly add power. We would like to specify that RNA-Seq was used as a screening procedure with primary focus on the expression of genes encoding for actin isoforms and ABPs, hypothesizing that aberrant actin would deregulate the expression of key binding proteins (see response 11 to Reviewer 2). The data from P4 and P5 were compared against the control separately and as a joined population. Both approaches resulted in the same list of aberrantly expressed ABPs. In our initial submission, we eliminated genes with low raw counts (< 300) to filter out ABPs where protein levels are not detectable. As we could not detect γ -SMA in P5, which had counts in the range of 1000, we believe that this is a valid and rational thresholding approach. The same genes were eliminated if we considered either the raw or normalized counts.

The actual expression of selected candidates was validated on the protein level in both patient fibroblasts and cells from a healthy control. The same cell culture was consistently used throughout all experiments within this project. For consistency, we decided not to include additional controls at this time. We do recognize that the informative value of such an approach is limited and results might be biased in favor of genes upregulated in patient samples, as mentioned by Reviewer 3. Therefore, we have removed the pathways analysis data and the discussion previously incorporated into the Supplementary Notes. We only show RNA-Seq data related to actin and ABPs, which we further validate at the protein level.

Reviewer #1 (Remarks to the Author):

The authors have adequately answered all my points. The electronic images they provide show platelets that are a bit activated as judged by the granule centralization, or even permeabilized. Nevertheless these images indeed suggest that ultrastructure of patient's platelets is comparable to the control, so I have no other question.

Reviewer #2 (Remarks to the Author):

The authors have adequately answered my queries on the original manuscript and I am happy to accept this manuscript.

Reviewer #3 (Remarks to the Author):

The authors' response is mostly satisfactory, but I have the following outstanding remarks.

Regarding GnomAD:

The authors state in their rebuttal: "Considering the mildness of the developmental disability and subclinical course of thrombocytopenia, we hypothesize that these rare variants might have been considered not significant, allowing the carriers to be included in the large sequencing cohorts that are incorporated into the GnomAD database."

Such an interpretation of the neurological phenotype does not justify calling the reported disease entity "Thrombocytopenia Microcephaly syndrome." It seems that the title does not describe the phenotype faithfully, and the microcephaly might be an over-interpretation. It may be best to call this disorder "Developmental disability with macrothrombocytopenia."

The possible presence of pathogenic variants in GnomAD should be referred to in the main text, alongside the text stating that the reported variants are absent from public databases (i.e., some of the contents of the Supplementary Note should be in the main text). The Supplementary Note itself should be referred to in the appropriate section in page 17.

Regarding the RNA-seq analysis:

I'm afraid the authors have failed to grasp and respond adequately to my criticisms. My comments are not related to the fact that "multiple biological replicates would significantly add power" at all (even though that is true and desirable). My concerns relate to problems with the experimental design / bioinformatics.

Let's recall that the authors have generated three technical replicates from each of two patients and three technical replicates from one control (I use the word "technical" here to refer to the use of a different flask of fibroblasts to generate a separate library for the same person; I use the word "biological" to refer to the use of a different person's fibroblasts). The authors state that "DESeq2 (v1.10.1) was used to find differentially expressed genes using standard parameters." This suggests that when the six patient samples were analysed together, they were treated as exchangeable. This is inappropriate because it ignores the heterogeneity in expression due to the grouping of samples by patient.

Secondly, comparing one sample with one other sample without biological replication is inappropriate for the reasons I stated previously: the differences will be driven largely by the fact that the comparison is between two different people (with substantial genetic differences other than in ACTB and differences in environmental exposures) rather than two conditions. Hence, the inference will tell you nothing meaningful about disease.

This is made obvious by the fact that when the authors compare one patient with one control, they find approximately 10k DEGs, but when they compare two patients with one control they only identify approximately 3k DEGs. Why are many thousands of DEGs from the single-patient comparison not being found when the two patients are grouped together? The answer must be that it is because most of those genes found do not vary because of disease but rather because of other factors. The claim that the more severely affected case having 10.7K rather than 8.3K DEGs in the less affected case compared to the control is "in line with the more severe phenotype" is unfounded. This could be for any number of reasons unrelated to disease, including genetic and environmental factors.

Other than generating a second set of control triplicates from a fourth individual (which would be the appropriate, albeit time-consuming, thing to do), I can suggest the following:

- 1) Seek a statistician's help to combine patient samples in a manner that accounts for the hierarchy statistically
- 2) Be very clear about the caveats of the analysis (due to a lack of biological replication) in the text and make clear that the RNA-seq results are suggestive / designed only to generate hypotheses rather than draw conclusions

Lastly, the text on RNA-seq is riddled with errors, lack of detail and inconsistencies:

- "A subset of 213 genes encoding for actin isoforms and known ABPs (based on 3) was assembled and analyzed further. Of the 207 ABPs assessed, ..." ; what happened to the remaining six genes? Also I suggest not using the word "assembled" because that has a specific, different meaning in this field.

- "Significantly upregulated and downregulated genes were chosen based on different absolute log₂ fold change values." - significance should be based on thresholding on p-value. Absolute log₂ fold changes cannot distinguish between upregulated and downregulated genes, for the obvious reason that the sign is lost. Furthermore, where the authors state "(DEG, Benjamini-Hochberg adjusted-value ≤ 0.01 , FC >1)," one wonders why the fold change for a gene to be declared a DEG needs to be >1 and can't be <1 (i.e. surely a DEG can be up-regulated as well as down-regulated).

- "(c) Heat map showing unsupervised clustering using the 500 genes with the highest variance in gene expression over all samples." The plot title states "124" not 500 genes and the two sets of patient samples do not cluster together! (the green group of samples is in the middle).

- "showing equal expression of both mutant and native ACTB mRNA in RNA-Seq data, indicates in red boxed region" - give number of reads supporting each allele and replace "indicates" with "indicated"

- "using feature Counts method" -> "using the featureCounts methods"

- "We revised this list by eliminating the 14 genes with lowest raw and normalized counts (Supplementary Table 6)" - at least stipulate the threshold (base mean < 150?)

- "Whilst whole exome sequencing shows a significant upregulation of ACTG2 mRNA" - I assume the authors mean "RNA-seq" or "whole transcriptome sequencing" not "whole exome sequencing"

- "Principal component analysis showed clear grouping of replicates for one individual" -> "...for each individual"

Reviewers' comments:

Reviewer #1 (Remarks to the Author):

The authors have adequately answered all my points. The electronic images they provide show platelets that are a bit activated as judged by the granule centralization, or even permeabilized. Nevertheless, these images indeed suggest that ultrastructure of patient's platelets is comparable to the control, so I have no other question.

Reviewer #2 (Remarks to the Author):

The authors have adequately answered my queries on the original manuscript and I am happy to accept this manuscript.

Reviewer #3 (Remarks to the Author):

The authors' response is mostly satisfactory, but I have the following outstanding remarks.

Regarding GnomAD:

The authors state in their rebuttal: "Considering the mildness of the developmental disability and subclinical course of thrombocytopenia, we hypothesize that these rare variants might have been considered not significant, allowing the carriers to be included in the large sequencing cohorts that are incorporated into the GnomAD database."

Such an interpretation of the neurological phenotype does not justify calling the reported disease entity "Thrombocytopenia Microcephaly syndrome." It seems that the title does not describe the phenotype faithfully, and the microcephaly might be an over-interpretation. It may be best to call this disorder "Developmental disability with macrothrombocytopenia."

We would like to clarify that a head circumference between 2-3 SD below the mean is usually defined as borderline/mild microcephaly. We termed the disease "Thrombocytopenia Microcephaly Syndrome" as thrombocyte number and head circumference are two parameters that can be objectively measured. We do not agree with the Reviewer's suggestion that this disorder should be called "Developmental disability and macrothrombocytopenia". The term "macrothrombocytopenia" does not faithfully describe the platelet anisotropy (variable size including normal and enlarged platelets) observed in these patients. Further, we believe that "developmental disability" is not specific enough to describe the patient phenotype and not applicable to all patients in our cohort. However, with this rationale, we see that we should not use the term microcephaly in the disease name (as patient 2 does not present with this phenotype). As such, we propose renaming the disease "*ACTB*-Associated Syndromic Thrombocytopenia (*ACTB*-AST)". The term syndromic indicates that these patients present with symptoms additional to thrombocytopenia. Accordingly, we have changed the title of the manuscript to "*ACTB*-Associated Syndromic Thrombocytopenia" and replaced *ACTB*-TMS with *ACTB*-AST in every instance throughout the main text, figure and supplementary files.

The possible presence of pathogenic variants in GnomAD should be referred to in the main text, alongside the text stating that the reported variants are absent from public databases (i.e., some of the contents of the Supplementary Note should be in the main text). The Supplementary Note itself should be referred to in the appropriate section in page 17.

The following sentence have been added to the main text:

- “The gnomAD database lists 7 additional heterozygous variants in the 3’ region of *ACTB* (Supplementary Table 9), each of which is seen in a single individual. It is not currently known whether these variants are truly benign, not fully penetrant, or causative for *ACTB*-AST (Supplementary Notes).” – Discussion on Page 13
- “All *ACTB* variants were absent from public databases (see also Supplementary Notes on 3’*ACTB* variants listed in the gnomAD database)” - Materials and Methods on Page 17.

Regarding the RNA-seq analysis:

I'm afraid the authors have failed to grasp and respond adequately to my criticisms. My comments are not related to the fact that "multiple biological replicates would significantly add power" at all (even though that is true and desirable). My concerns relate to problems with the experimental design / bioinformatics.

Let's recall that the authors have generated three technical replicates from each of two patients and three technical replicates from one control (I use the word "technical" here to refer to the use of a different flask of fibroblasts to generate a separate library for the same person; I use the word "biological" to refer to the use of a different person's fibroblasts). The authors state that "DESeq2 (v1.10.1) was used to find differentially expressed genes using standard parameters." This suggests that when the six patient samples were analysed together, they were treated as exchangeable. This is inappropriate because it ignores the heterogeneity in expression due to the grouping of samples by patient.

Secondly, comparing one sample with one other sample without biological replication is inappropriate for the reasons I stated previously: the differences will be driven largely by the fact that the comparison is between two different people (with substantial genetic differences other than in *ACTB* and differences in environmental exposures) rather than two conditions. Hence, the inference will tell you nothing meaningful about disease.

This is made obvious by the fact that when the authors compare one patient with one control, they find approximately 10k DEGs, but when they compare two patients with one control they only identify approximately 3k DEGs. Why are many thousands of DEGs from the single-patient comparison not being found when the two patients are grouped together? The answer must be that it is because most of those genes found do not vary because of disease but rather because of other factors. The claim that the more severely affected case having 10.7K rather than 8.3K DEGs in the less affected case compared to the control is "is in line with the more severe phenotype" is unfounded. This could be for any number of reasons unrelated to disease, including genetic and environmental factors.

Other than generating a second set of control triplicates from a fourth individual (which would be the appropriate, albeit time-consuming, thing to do), I can suggest the following:

- 1) Seek a statistician's help to combine patient samples in a manner that accounts for the hierarchy statistically
- 2) Be very clear about the caveats of the analysis (due to a lack of biological replication) in the text and make clear that the RNA-seq results are suggestive / designed only to generate hypotheses rather than draw conclusions

The reviewer has properly criticized the limitations of our initial RNA-Seq design and the caveats of our analysis. We appreciate their comments and apologize that our previous response did not adequately address their concerns.

Regarding the “technical” vs “biological” replicate, we have included the following sentence to specify exactly what we mean by triplicate in the materials and methods section of the text – “All experiments were performed in triplicate, meaning that RNA was independently extracted three times from each culture”.

Regarding the identification of differentially expressed genes with DESeq2, we performed three separate analyses: 1) P4 vs control (Suppl. Table 2), 2) P5 vs control (Suppl. Table 3), and 3) P4 and P5 combined vs control (Suppl. Table 4). To specify this better in the text we have now included the sentence – “Triplicates from P4 and P5 were compared against control triplicates separately and as a combined group (P4 and P5) against the control (Supplementary Tables 2-4)”.

We have refined the list of significantly deregulated ABP (Suppl. Table 6 and Figure 5a) to include only those genes that appear on all three lists. This does not affect the five ABP candidates that we assessed further. Due to this refinement, the following sections of the manuscript have been modified:

- “Of the 207 ABPs genes assessed, 44 were found to be significantly deregulated in all three analyses (P4 vs control, P5 vs control and *ACTB*-AST patients combined vs control). We revised this list by eliminating the 7 genes with the lowest raw and normalized counts (base mean < 150; Supplementary Table 6)” – Page 20 (main text).
- The lists of genes given in Suppl. Table 6 and Fig. 5a have been shortened accordingly.

As we described in the previous rebuttal, we used this method to pre-screen the actin isoforms and generate a list of potentially affected ABPs. In the end, candidate genes were co-selected from literature searches, and were each assessed at the protein level in fibroblasts and thrombocytes. We agree that further conclusions are unable to be drawn from the transcriptome analysis in its current state without biological replication. Accordingly, the following amendments have been made:

- We have removed all statements about a relationship between the number of DEGs and patient severity in the Supplementary Notes.
- We have removed the clustered analysis previously given as Supplementary Figures 4a and 4c and all related text. We still include the principal component analysis to visually illustrate the reproducibility of our replicates and show variation between samples.
- We have added the following statement to text:
 - “Whole-transcriptome sequencing experiments were designed and performed to screen actin isoform transcription (including that of the mutated *ACTB* allele) and to select a subset of ABP's for further analysis at the protein level. Further conclusions about the impact of these *ACTB* mutations are unable to be drawn due to the lack of additional healthy control biological replicates” – Page 20 (main text).
 - “The global gene expression profile of P4, P5 and healthy control fibroblasts was assessed to evaluate actin isoform transcription and to define a shortlist of affected ABP's, which were subsequently examined at the protein level” – Page 1 (Suppl. Notes).
 - “Further conclusions about the impact of these *ACTB* mutations are unable to be drawn from the transcriptome analysis due to the lack of additional healthy control biological replications. Additional studies are needed to assess the impact of *ACTB*-AST mutations on the transcriptome” – Page 2 (Suppl. Notes).

Lastly, the text on RNA-seq is riddled with errors, lack of detail and inconsistencies:

- "A subset of 213 genes encoding for actin isoforms and known ABPs (based on 3) was assembled and analyzed further. Of the 207 ABPs assessed, ..."; what happened to the remaining six genes? Also, I suggest not using the word "assembled" because that has a specific, different meaning in this field.

The remaining six genes encode for actin isoforms and we have clarified this in the text and in Supplementary Table 5. The word "assembled" has been replaced with compiled. The new text reads - "A subset of 213 genes encoding for actin isoforms (6 genes) and known ABPs (207 genes, based on ³) was compiled and analyzed further" (Page 20).

- "Significantly upregulated and downregulated genes were chosen based on different absolute log₂ fold change values." - significance should be based on thresholding on p-value. Absolute log₂ fold changes cannot distinguish between upregulated and downregulated genes, for the obvious reason that the sign is lost. Furthermore, where the authors state "(DEG, Benjamini-Hochberg adjusted-value ≤ 0.01 , FC >1)," one wonders why the fold change for a gene to be declared a DEG needs to be >1 and can't be <1 (i.e. surely a DEG can be up-regulated as well as down-regulated).

We have corrected the text as suggested and indicated that the adjusted p-value ≤ 0.01 was used as a threshold for identifying significantly deregulated genes. We have removed the >1 or <1 from the text. The new text reads: "The differentially expressed genes were filtered using the adjusted p-value ≤ 0.05 . Significantly upregulated and downregulated genes were chosen based on the Benjamini-Hochberg adjusted P-value ≤ 0.01 " (Page 20 of main text).

- "(c) Heat map showing unsupervised clustering using the 500 genes with the highest variance in gene expression over all samples." The plot title states "124" not 500 genes and the two sets of patient samples do not cluster together! (the green group of samples is in the middle).

We have removed the clustered analysis originally given as Supplementary Figures 4a and 4c. The figure and legend now just refer to the principal component analysis, which we continue to include as a visual representation of our data. The new figure legend reads: "**Supplementary Figure 4. Whole-transcriptome analysis of control, P4 and P5 primary dermal fibroblasts.** Plot of principal component analysis (PCA) illustrating the grouping of replicates and variation between the different fibroblast cultures. Control fibroblasts (culture ID 31660) are shown in blue, P4 fibroblasts (culture ID 34012) are shown in red and P5 fibroblasts (culture ID 30563) are shown in green." (Page 10 of Suppl. Notes).

- "showing equal expression of both mutant and native ACTB mRNA in RNA-Seq data, indicates in red boxed region" - give number of reads supporting each allele and replace "indicates" with "indicated"

We have corrected the text and provided the read numbers in the legend of Supplementary Figure 5. The new text now reads: "showing expression of both mutant and native *ACTB* mRNA (red boxed regions) in patient 4 (P4 - p.Ala331Valfs27: 40% reads supporting deletion - reference read count 2413-1798, read count with deletion 1257-2096) and patient 5 (P5 - p.Ser338_Ile341del: 44% reads supporting deletion - reference reads 1877-1943, read count with deletion 1292-1662) fibroblasts, as determined by RNA-Seq;" (Page 11 of Supplementary Notes).

- "using feature Counts method" -> "using the featureCounts methods"

This has been corrected as "featureCounts function" (Page 19 main text).

- "We revised this list by eliminating the 14 genes with lowest raw and normalized counts (Supplementary Table 6)" - at least stipulate the threshold (base mean < 150?)

We have specified that the cut-off threshold was base mean < 150 (Page 20 main text).

- "Whilst whole exome sequencing shows a significant upregulation of ACTG2 mRNA" - I assume the authors mean "RNA-seq" or "whole transcriptome sequencing" not "whole exome sequencing"

We have corrected this typo to 'whole transcriptome sequencing' (Page 8 main text).

- "Principal component analysis showed clear grouping of replicates for one individual" -> "...for each individual"

We have corrected this text accordingly to 'for each individual' (Page 2 Suppl. Notes).

In addition to the changes requested by Reviewer 3, we have carefully corrected a number of small points throughout the text files and we have formatted the manuscript in accordance with the journals requirements. Please see the list of additional changes below.

Additional changes include:

- "and show that variant transcripts account for 50% of the *ACTB* mRNA present in these cells" – deleted from Page 7 (main text).
- "P4 cell" → "P4 cells" on Page 9 (main text).
- "Fig. 5a" → "Fig. 5a, red boxes" on Page 9 (main text).
- "Supporting loop" corrected to "Supporting-loop" throughout the text.
- "disease-associated *ACTB* p.Glu364Lys mutation" → "disease-associated *ACTB* p.Glu364Lys and p.Arg183Trp mutations" on Page 10 (main text).
- "*ACTB*-TMS-associated thrombocytopenia" → "thrombocytopenia in *ACTB*-AST patient cells" on Page 15 (main text).
- "recruited into mutant β -CYA bundles" → "recruited to mutant β -CYA bundles" on Page 16 (main text).
- "density of 2.5×10^3 " → "density of 2.5×10^3 " on Page 19 (main text).
- "For library preparations, TruSeq Stranded" → "For library preparations, the TruSeq Stranded" on Page 19 (main text).
- "reads are aligned" → "reads were aligned" on Page 19 (main text).
- "samples are merged and used as guide" → "samples were merged and used as a guide" on Page 19 (main text).
- "Each protein of interest" → "For quantification, each protein of interest" on Page 21 (main text).
- "A P902 electron microscope (Zeiss, Oberkochen, Germany) was used to analyze the specimens" → "A EM 902 electron microscope (Zeiss, Oberkochen, Germany) was used to analyze the specimens at 80 kV" on Page 24 (main text).
- "effected residues" → "impacted residues" on Page 32 (main text).
- "4 experiments given" → "4 experiments is given" on Page 33 (main text).
- "(b) the binding" → "(b) The binding" on Page 35 (main text).
- "Closeups" → "Close-ups" on Page 35 (main text).
- "supporting loop" → "supporting-loop" on Page 35 (main text).
- ", yet comparable incorporation of intracellular organelles." – deleted from Page 35 (main text).
- "The organization and integrity of β -tubulin" → "Microtubule organization and integrity" on Page 36 (main text).
- "labelled MK" → "labelled MKs" on Page 36 (main text).

- “ β -tubulin is” → “microtubules are” on Page 36 (main text).
- “disordered β -tubulin” → “disordered microtubules” on Page 36 (main text).
- “disorder microtubule” → “disordered microtubule” on Page 36 (main text).
- “each of the variants has only been seen” → “each has only been seen” on Page 1 (Suppl. Notes).
- “2912” → “2307” on Page 2 (Suppl. Notes).
- “CM loop” corrected to “CM-loop”
- “showing equal expression” to “showing expression” on Page 11 (Suppl. Notes).

Formatting changes:

- Author postal addresses added.
- Supplementary Fig. 6 and Supplementary Fig. 8 have been added as the journal requires original western blot images for portioned blots given in the main figures. Accordingly, the supplementary figures have been renumbered from Supplementary Figure 6 onwards. Additionally, a sentence clarifying our methodology has been added to the materials and methods section on Page 21 of the main text – “Based on the Ponceau S staining and pre-stained marker, membranes were cut into 2-4 segments and probed with antibodies for proteins running in the respective size range (Supplementary Fig. 6 and Supplementary Fig. 8)”.
- Subheadings have been shortened to <60 characters (incl. spaces).
- Scale bar measurements added to each respective legend and removed from figures.
- Errors corrected (s.d. and s.e.m.).
- Sub-panels in Figure 3 renumbered.

REVIEWERS' COMMENTS:

Reviewer #3 (Remarks to the Author):

I have no further comments.